# Structural basis for excitatory neuropeptide signaling

Valeria Kalienkova ●[1,4], Mowgli Dandamudi[2], Cristina Paulino ●[1,3] ✉ &
Timothy Lynagh ●[2] ✉

Rapid signaling between neurons is mediated by ligand-gated ion channels, cell-surface proteins with an extracellular ligand-binding domain and a membrane-spanning ion channel domain. The degenerin/epithelial sodium channel (DEG/ENaC) superfamily is diverse in terms of its gating stimuli, with some DEG/ENaCs gated by neuropeptides, and others gated by pH, mechanical force or enzymatic activity. The mechanism by which ligands bind to and activate DEG/ENaCs is poorly understood. Here we dissected the structural basis for neuropeptide-gated activity of a neuropeptide-gated DEG/ENaC, FMRFamide-gated sodium channel 1 (FaNaC1) from the annelid worm *Malacoceros fuliginosus*, using cryo-electron microscopy. Structures of FaNaC1 in the ligand-free resting state and in several ligand-bound states reveal the ligand-binding site and capture the ligand-induced conformational changes of channel gating, which we verified with complementary mutagenesis experiments. Our results illuminate channel gating in DEG/ENaCs and offer a structural template for experimental dissection of channel pharmacology and ion conduction.

Ligand-gated ion channels (LGICs) are cell membrane proteins that convert extracellular chemical signals into transmembrane ionic current, thus contributing to rapid inter-cellular signaling and chemo-sensation[1,2]. Major LGIC superfamilies, such as nicotinic receptors and ionotropic glutamate receptors, are found in prokaryotes and eukaryotes, and are gated by small amino acid or biogenic amine ligands[1,3]. This contrasts with a third LGIC superfamily that is more specific to animals and close relatives, the trimeric degenerin/epithelial sodium channels (DEG/ENaCs)[2,4–7]. Despite having arisen relatively recently, DEG/ENaCs are diverse in terms of gating stimuli, as the superfamily includes constitutively active channels, pH-gated channels, osmolarity-gated channels, mechanically gated channels and neuropeptide-gated channels, among others[2,4]. DEG/ENaCs are often expressed in neurons, where their gating causes depolarization due to selective cation permeability[8,9], but are also expressed in numerous other cells, such as muscle and epithelia[2,10].

The fact that such diverse stimuli activate DEG/ENaCs raises several questions, ranging from evolutionary to physiological to biophysical. For example, did sensitivity to different ligands emerge independently and on demand in different animal lineages? And from a biophysical perspective, is there a gating machinery unique to the DEG/ENaC architecture that converts very different biophysical stimuli into similar conformational change at the channel gate? So far, our knowledge of DEG/ENaC channel architecture and gating derives mostly from X-ray or cryo-electron microscopy (cryo-EM) data[11,12] and complementary biophysical experiments[13] on vertebrate acid-sensing ion channels (ASICs), a family of proton-gated DEG/ENaCs. These, together with recent structures of the ENaC extracellular domain, show that DEG/ENaCs are assembled by three homologous subunits, each with a channel-forming transmembrane domain and a large extracellular domain, in threefold symmetry around a central pore[14,15]. As inferred from high-resolution structures of chicken ASIC1, channel gating involves the following conformational changes in each subunit. The protonation of numerous side chains leads to the collapse of a large part of the extracellular domain, whereby the mid-peripheral domain ('thumb') is drawn upward toward the upper,

[1]Groningen Biomolecular Sciences and Biotechnology Institute, University of Groningen, Groningen, the Netherlands. [2]Michael Sars Centre, University of Bergen, Bergen, Norway. [3]Biochemistry Center, Heidelberg University, Heidelberg, Germany. [4]Present address: Department of Biomedicine, University of Bergen, Bergen, Norway. ✉e-mail: cristina.paulino@bzh.uni-heidelberg.de; tim.lynagh@uib.no

'finger' domain; concomitantly, β-strands of the low-peripheral 'palm' and 'wrist' domains move outward, pulling channel-forming α-helices peripherally to open the channel[12,14].

It is unknown whether protons and larger, more canonical transmitters such as neuropeptides induce the same biophysical mechanism of channel gating in cognate DEG/ENaCs. The relationship between neuropeptide-gated and other DEG/ENaCs is also interesting from an evolutionary perspective, as neuropeptide-based, paracrine signaling systems potentially predated and gave rise to more modern synaptic systems[16,17], and neuropeptide-gated DEG/ENaCs occur in distinct animal types that diverged a long time ago[18,19]. This raises the possibility that neuropeptide-gated channels constitute one of the earliest occurring DEG/ENaCs and that understanding their ligand-induced gating may offer broad insights into mechanisms of DEG/ENaC function. Two distinct families of neuropeptide-gated DEG/ENaCs have been described so far. These include, from radially symmetric hydrozoans, the hetero-trimeric pyroQWLGGRFamide-gated Na⁺ channels (HyNaCs)[20], and from bilaterally symmetric mollusks and annelids, the homo-trimeric FMRFamide-gated Na⁺ channels (FaNaCs)[19,21]. The short neuropeptide FMRFamide (H–Phe–Met–Arg–Phe–$NH_2$, 'FMRFa') is of particular importance in bilaterian animals, as its broad neural expression makes it a common marker of the nervous system in numerous model invertebrates, in which it mediates signaling via FaNaCs and/or G-protein-coupled receptors[22].

In this Article, to uncover the structural basis for excitatory neuropeptide activity and establish principles of ligand recognition and channel gating in the DEG/ENaC superfamily, we have investigated the structure of FaNaC1, an FMRFa-gated DEG/ENaC from the annelid *Malacoceros fuliginosus*, using cryo-EM. We solved high-resolution structures of FaNaC1 alone, with full agonist FMRFa, with partial agonist ASSFVRIa, and with both FMRFa and pore-blocker diminazene, identifying the ligand-binding site and elucidating the conformational changes induced by ligand binding. Together with complementary mutagenesis and electrophysiological experiments, these results establish ligand recognition and channel gating mechanisms and offer a structural template for the experimental dissection of function throughout the DEG/ENaC channel superfamily.

## Results

### FaNaC structural architecture
To investigate the structure of neuropeptide-gated DEG/ENaC channels, we utilized FaNaC1 from the annelid *Malacoceros fuliginosus*[21], a channel with typical FMRFa-gated Na⁺-selective currents (Fig. 1a), that expressed well in preliminary screening. We transduced human embryonic kidney 293S (HEK293S) cells, purified FaNaC1, and incorporated it into lipid nanodiscs for subsequent cryo-EM study. Preparations of FaNaC1 alone and with FMRFa (30 μM) yielded 3D reconstructions with a global resolution of 2.7 and 2.5 Å, respectively (Fig. 1b,c, Table 1 and Extended Data Figs. 1 and 2). For both structures, density could be unambiguously assigned to amino acid sequence based on mostly continuous main chain density and numerous distinctive side chain densities (Extended Data Figs. 1 and 2). The 63-amino-acid C-terminal tail was not resolved. This domain is highly variable across DEG/ENaCs, and experiments with ASICs suggest it is flexible and largely dispensable[14,23,24].

The structures confirm that FaNaCs are trimeric like other DEG/ENaCs[14,15], with three subunits forming a central channel pore (Fig. 1b,c). Each subunit comprises a minimal N terminus and a nonresolved 63-amino-acid C terminus facing the intracellular side, two transmembrane segments (TM1 and TM2), and a large extracellular domain (Fig. 1d,e). The extracellular domain can be divided into palm, thumb, finger and knuckle domains similar to those previously described for ASIC[14] and ENaC[15] (Fig. 1e). TM2 is unwound at a GIS motif in the middle of the membrane, yielding discontinuous upper 'TM2a' and lower

'TM2b' α-helical segments within each subunit (Fig. 1e and Extended Data Figs. 1–3). Consequently, TM2a from one subunit essentially forms a membrane-spanning helix with TM2b from the adjacent subunit (Fig. 1e). The upper, middle and lower segments of the channel pore are lined by TM2a, the GIS motif and a re-entrant loop from the short pre-TM1 N-terminal segment, respectively (Fig. 1e). Thus, the channel architecture of FaNaC1 is similar to that of its distant DEG/ENaC cousin, ASIC1 (refs. 11,25), suggesting that this architecture is probably adopted by most channels of the DEG/ENaC superfamily. We also observe that the hydrophobic periphery of the channel is thinner than the membrane bilayer, such that the outer leaflet of the membrane bends to make way for hydrophilic, lateral fenestrations between adjacent subunits, possibly creating a path for water and ions into the channel pore (Extended Data Fig. 4).

### Ligand-binding site
In the FaNaC1/FMRFa structure, we observed a discrete cryo-EM density in a small pocket at the upper corner of each subunit, which fits a single FMRFa molecule (Fig. 2a and Extended Data Fig. 2). The FMRFa-binding pocket is formed by α-helical segments α1–α3a (residues V87 to F144), the β6–β7 loop of the same subunit (residues D234 to G241), and partly by α6 from the adjacent subunit (G423–K428, Fig. 2a). The basis for ligand recognition appears to be mostly hydrophobic interactions. The FMRFa N-terminal phenylalanine residue (F1) is positioned near the entrance to the pocket and the M2 side chain orients downward between α2-F129, β6–β7 loop-I236 and M238, and α6-F431. FMRFa R3 orients upward: the density for the guanidino moiety in our map is relatively weak, but the modeled side chain is 4–6 Å from polar side chains α1-D101 and β6–β7 loop-E235 and R237. Finally, FMRFa F4 and C-terminal amide sit deep in the pocket, with the F4 side chain surrounded as closely as 3.5–4.6 Å by the hydrophobic side chains of α1-F97, P103, α2-V122, A126 and F129, and β6–β7 loop-P240. Additionally, FMRFa M2 main chain carbonyl oxygens are 2.5–3.1 Å from the FaNaC1 α2-Q133 amide side chain (Fig. 2a), indicative of a potential polar interaction.

To establish the functional importance of such interactions, we compared FMRFa potency at wild-type (WT) and 18 mutant channels with amino acid substitutions at selected positions, via heterologous expression in *Xenopus laevis* oocytes and two-electrode voltage clamp (Fig. 2b,c). We measured potency by establishing the half-maximal effective concentration ($EC_{50}$) of FMRFa during the increasing part of the concentration–response relationship, excluding a decrease in current amplitudes with higher FMRFa concentrations (Fig. 2b). Additional experiments suggested this decrease is due to slow recovery from desensitization with high concentrations of the ligand (Extended Data Fig. 5a,b). As this decrease also occurs at positive membrane potentials and with other, uncharged peptide analogs (Extended Data Fig. 5c,d), we think it is not a sign of channel block by high concentrations FMRFa, which has been suggested for mollusk FaNaCs[26].

Mutations in the α2 helix had the largest effects on FMRFa potency, with F129L and F129A mutations decreasing potency 10- to 20-fold ($EC_{50}$: WT, 850 ± 270 nM; F129L, 9 ± 3 μM; F129A, 18 ± 8 μM; each *n* = 4) and F129Q decreasing potency ~100-fold (Fig. 2b,c), suggesting that van der Waals interactions between FaNaC1 F129 and FMRFa M2 and F4 contribute substantially to FMRFa binding. Whereas V122F and V122Q mutations had no effect on FMRFa potency, V122A caused a 20-fold increase in FMRFa potency (Fig. 2b,c; $EC_{50}$ 40 ± 10 nM, *n* = 4). The putative polar interactions we probed via mutagenesis make, at most, relatively subtle contributions to FMRFa binding. For example, Q133N and Q133L mutations, essentially retracting or removing a hydrogen bond partner from the FMRFa main chain, and D101A/E235A, removing two oppositely charged binding partners of the FMRFa R3 side chain, caused two- to fourfold decreases in potency (Fig. 2c). Finally, on the inner wall of the binding pocket, M238 and F431 mutations also had

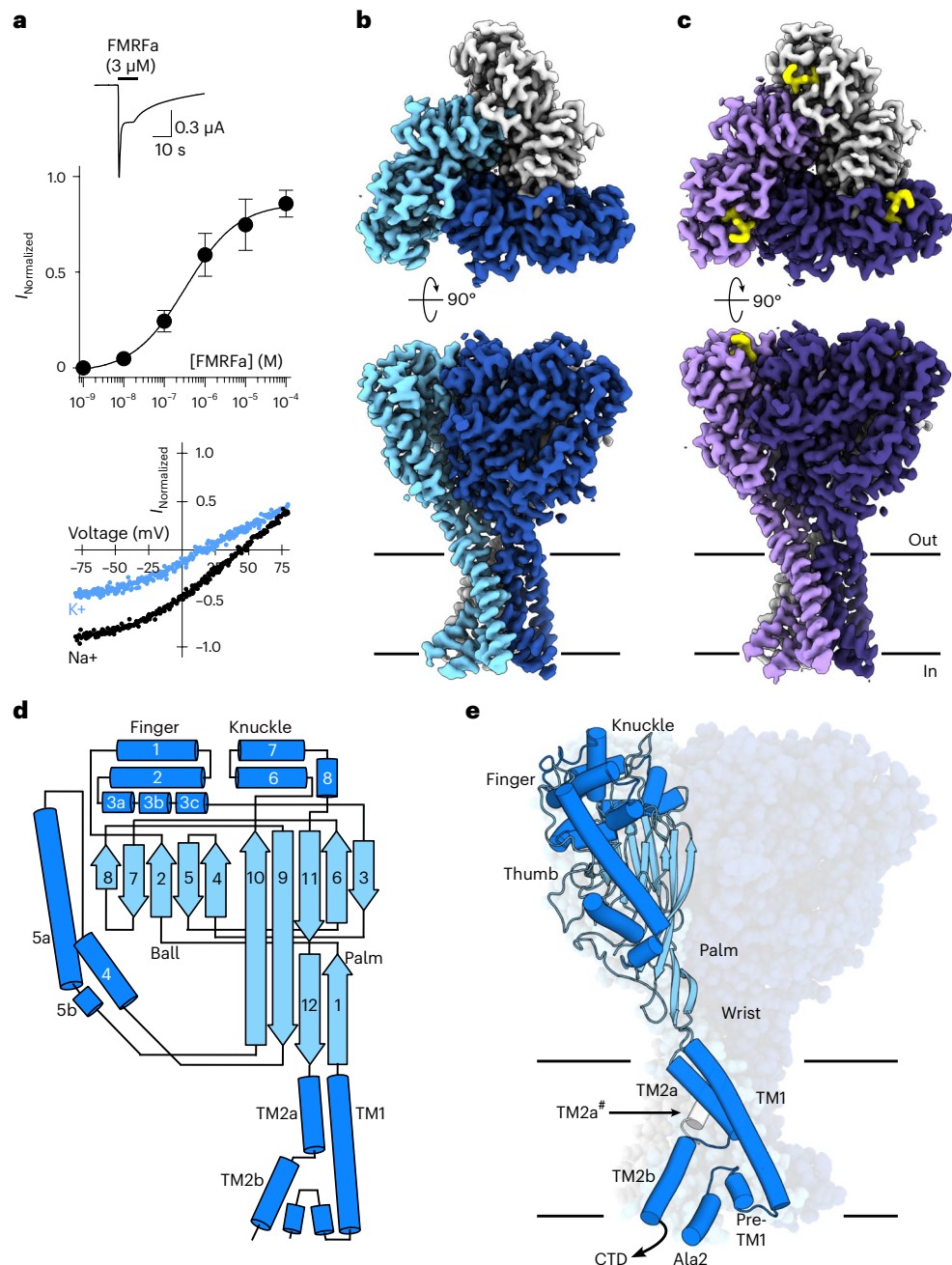

**Fig. 1 | *Malacoceros fuliginosus* FaNaC1 architecture. a**, Top, example two-electrode voltage clamp current in a FaNaC1-expressing oocyte in response to FMRFa (application indicated by black bar). Middle, mean concentration-dependent FMRFa-gated current amplitudes normalized to maximum (mean ± s.e.m., $n$ = 4 oocytes). Bottom, FMRFa-gated (3 μM) current amplitude at different oocyte membrane voltages in extracellular Na+- or K+-based solutions. **b,c**, Cryo-EM density maps of FaNaC1 (subunits in blues) (**b**) and FaNaC1/FMRFa (subunits in purples, FMRFa in yellow) (**c**), viewed from the extracellular side (top) and from within the lipid bilayer (bottom). Lines indicate bilayer. **d**, Schematic of major secondary structure elements. α-helices, dark blue; β-strands, light blue. **e**, FaNaC1 model highlighting one subunit, colored as in **d**. Ala2, amino acid residue immediately following starting methionine. CTD, C-terminal tail, not resolved; TM2a#, adjacent subunit.

little if any effect on FMRFa potency (Fig. 2c). Thus, FMRFa binding seems to rely mostly on hydrophobic interactions with side chains of the α2 helix.

We further validated the FMRFa binding mode in our structures by testing the potency of analogous peptides differing from FMRFa only at the F1, R3 or F4 position. We replaced R3 with citrulline, isosteric but neutral, or glutamine, shorter and neutral, yielding FMρFa and FMQFa, respectively. Compared to FMRFa, FMρFa was equally potent (EC$_{50}$ 690 ± nM, $n$ = 4) and FMQFa was only fivefold less potent (EC$_{50}$ 4.5 ± 2 μM, $n$ = 4, Fig. 2d). Thus, polar interactions between FMRFa R3

and FaNaC1 D101/E235 are relatively dispensable and FMRFa R3, sitting at the upper/outer part of the binding site, contributes little to potency. Removing the N-terminal F1 side chain also had relatively little effect, with AMRFa showing slightly increased potency compared to FMRFa (EC$_{50}$ 300 ± 150 nM, $n$ = 3). In contrast, removing the C-terminal F4 side chain drastically reduced potency, with FMRAa barely activating detectable currents at 100 μM (Fig. 2d and Extended Data Fig. 5d). This supports our structural results and shows that potency derives primarily from FMRFa F4, and potentially M2, engaging FaNaC1 hydrophobic side chains.

**Table 1 | Cryo-EM data collection, refinement and validation statistics**

| | Apo (EMDB-16982), (PDB 8ON8) | FMRFa (EMDB-16981), (PDB 8ON7) | ASSFVRIa (EMDB-16983), (PDB 8ON9) | FMRFa/dim. (EMDB-16984), (PDB 8ONA) |
|---|---|---|---|---|
| **Data collection and processing** | | | | |
| Magnification | 49,407 | 49,407 | 49,407 | 49,407 |
| Voltage (kV) | 200 | 200 | 200 | 200 |
| Electron exposure (e⁻ Å⁻²) | 47.47 | 50.11 | 47.40 | 47.20 |
| Defocus range (µm) | −0.3 to −2.0 | −0.3 to −2.0 | −0.3 to −2.0 | −0.3 to −2.0 |
| Pixel size (Å) | 1.022 | 1.022 | 1.022 | 1.022 |
| Symmetry imposed | C3 | C3 | C3 | C3 |
| Initial particle images (no.) | 2,659,492 | 3,235,905 | 4,662,322 | 3,893,627 |
| Final particle images (no.) | 376,869 | 458,934 | 477,006 | 137,558 |
| Map resolution (Å) | | | | |
| FSC threshold 0.143 | 2.67 | 2.52 | 2.39 | 2.96 |
| Map resolution range (Å) | 2.6–3.5 | 2.4–3.6 | 2.3–3.5 | 2.9–3.8 |
| **Refinement** | | | | |
| Initial model used (PDB code) | 8ON7 | AlphaFold | 8ON7 | 8ON7 |
| Model resolution (Å) | | | | |
| FSC threshold 0.143 | 2.4 | 2.2 | 2.2 | 2.7 |
| Model resolution range (Å) | 2.4–3.5 | 2.2–3.6 | 2.2–3.5 | 2.7–3.8 |
| Map sharpening $B$ factor (Å²) | N/A | N/A | N/A | N/A |
| $Q$-score | 0.49 | 0.55 | 0.54 | 0.50 |
| Model composition | | | | |
| Nonhydrogen atoms | 12,891 | 13,032 | 13,014 | 12,900 |
| Protein residues | 1,572 | 1,587 | 1,587 | 1,581 |
| Ligands | 21 | 24 | 24 | 18 |
| $B$ factors (Å²) | | | | |
| Protein | 123.38 | 104.22 | 107.97 | 123.56 |
| Ligand | 153.47 | 134.47 | 142.07 | 143.51 |
| Root mean square deviations | | | | |
| Bond lengths (Å) | 0.003 | 0.003 | 0.004 | 0.004 |
| Bond angles (°) | 0.630 | 0.687 | 0.725 | 0.720 |
| Validation | | | | |
| MolProbity score | 1.06 | 1.2 | 1.12 | 1.27 |
| Clashscore | 3.01 | 3.48 | 3.29 | 3.79 |
| Poor rotamers (%) | 0.2 | 0.2 | 0.2 | 0.6 |
| Rama Z | | | | |
| Whole | 1.34 | 1.48 | 1.48 | 0.59 |
| Helix | 1.25 | 1.27 | 1.49 | 0.16 |
| Sheet | 1.62 | 1.7 | 1.74 | 1.09 |
| Loop | 0.24 | 0.51 | 0.15 | 0.59 |
| Ramachandran plot | | | | |
| Favored (%) | 98.08 | 97.71 | 98.6 | 97.5 |
| Allowed (%) | 1.92 | 2.29 | 1.4 | 2.5 |
| Disallowed (%) | 0 | 0 | 0 | 0 |

N/A, not applicable; FSC, Fourier shell correlation.

## Divergence of FaNaCs from other DEG/ENaCs

We questioned the divergence of neuropeptide-gated FaNaCs from other DEG/ENaCs by examining this pocket in high-resolution structures of vertebrate ENaC and ASIC. Although α6 lies in a similar position in each channel, α1–α3 arrangement is vastly different in FaNaC1, ENaC and ASIC (Extended Data Fig. 6). Consequently, the FMRFa site is essentially obscured by an α-helix in ENaC and by a loop in ASIC. This suggests that the FMRFa-binding pocket is unique to the FaNaC family, and the enhancement of proton-gated currents that FMRFa elicits in ASICs[27] probably derives from binding to elsewhere on the channel, consistent with the central, extracellular vestibule peptide-binding site proposed by others for ASICs[28,29].

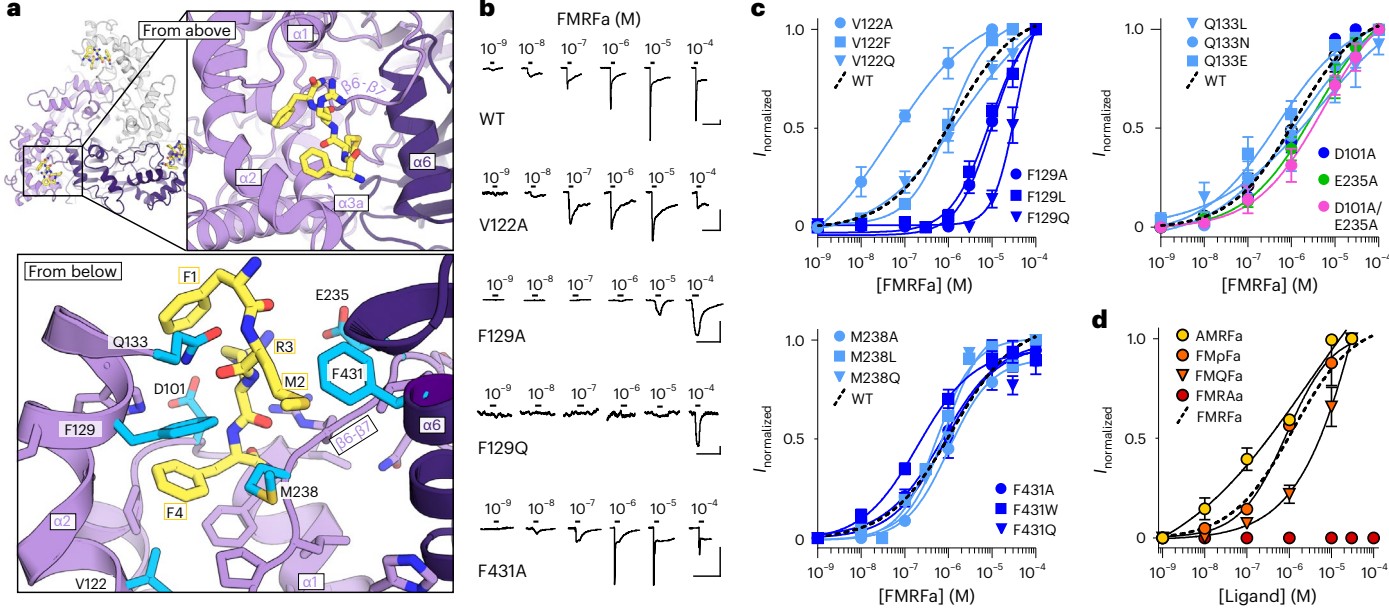

**Fig. 2 | FMRFa binding site. a**, Ligand-binding site in FaNaC1/FMRFa structure, viewed from the extracellular space ('from above', top) and from below the FMRFa-binding site (bottom). Subunits shaded in purples, FMRFa yellow. Residues tested via mutagenesis cyan and labeled. **b**, Example FMRFa-gated currents in oocytes expressing indicated WT and mutant channels. Scale bars, *x*, 30 s; *y*, 1 µA (WT, V122A and F129A) or 200 nA (F129Q and F431A). **c**, Mean (± s.e.m., *n* = 4 oocytes) normalized currents in response to increasing

FMRFa concentrations at indicated mutants. Data points beyond saturating concentrations omitted for clarity. WT curve repeated from Fig. 1a. **d**, Mean (± s.e.m., *n* = 3 oocytes for AMRFa and 4 oocytes for all other peptides) normalized currents in response to increasing concentrations of different ligands at WT FaNaC1. Data points beyond saturating concentrations omitted for clarity. WT curve repeated from Fig. 1a.

More surprisingly, we notice substantial amino acid sequence divergence in the finger domain even within the FaNaC family. Although α2-V122 is arguably somewhat conserved in various FaNaCs (valine in FaNaC1; valine, threonine or isoleucine in most other FaNaCs), other α1, α2 and α3 residues are difficult to align, even including α2-F129, the most influential residue for FMRFa potency in our experiments (Extended Data Fig. 7). Despite such different sequences here, annelid and mollusk FaNaCs arrive at an architecturally similar FMRFa binding site, as revealed by the comparative analysis of our structures and recently published structures of a FaNaC from the mollusk *Aplysia californica*[30]. Finger domains of both *Malacoceros* FaNaC1 and *Aplysia* FaNaC comprise relatively vertical α1 helices, horizontal α2 and α3a-b helices, and the β6–β7 loop, next to a horizontal α6 (or equivalent α8) helix from the adjacent subunit (Extended Data Fig. 7a). Although FMRFa binds in the same site in both channels, the peptide sits 'horizontally' in *Malacoceros* FaNaC1, with F4 orienting deeply in the pocket toward α2-V122 (Fig. 2a), but 'vertically' in *Aplysia* FaNaC, with F1 oriented most deeply into the pocket toward α3b-F188 (Extended Data Fig. 7a)[30].

**Contributions to FMRFa activity by finger and palm domains**
We had previously proposed an FMRFa-binding site ~20 Å away from this site, at the interface of adjacent subunits' β-ball and palm domains (orange in Fig. 3a and Extended Data Fig. 7) based on amino acid sequence analysis and severe effects of mutations in this site in several FaNaCs[21]. Our high-resolution maps seem to disprove that, motivating us to experimentally compare the finger site and palm/β-ball site. To this end, we compared the effects of modification of introduced cysteine residues in both of the sites by 2-(trimethylammonium)ethyl]methanethiosulfonate (MTSET), offering a readout of steric modification of the sites in 'real time'. In the finger domain, MTSET modification of α1-F97C, β6/β7-M238C and α2-F129C reduced FMRFa-gated current amplitude to approximately half in each case (Fig. 3a,b). In the palm domain, MTSET modification reduced β9-S282C currents to about half

but had no effect on the β11-N475C mutant (Fig. 3a,b). This shows that both sites can be modulated by MTSET.

Differences between the sites emerged when we compared the effects of MTSET on FMRFa potency (Fig. 3c,d). MTSET modulation decreased FMRFa potency via each of the finger domain cysteine residues, shifting EC$_{50}$ values from 250 ± 80 nM to 4 ± 2 µM at F97C, 5 ± 1 µM to >10 µM at F129C, and 140 ± 70 nM to 580 ± 290 nM at M238C (each *n* = 3, Fig. 3c). In contrast, FMRFa EC$_{50}$ values were either unchanged or slightly decreased at palm domain S282C (360 ± 100 nM and 170 ± 40 nM) and N475C (230 ± 110 nM and 140 ± 70 nM, each *n* = 3, Fig. 3d). Thus, the real-time addition of bulk to the finger domain site decreases FMRF-gated currents because of a decrease in FMRFa potency, presumably by impairing FMRFa binding. In contrast, addition of bulk to the palm/β-ball site impairs FMRFa-gated currents without a decrease in potency, presumably by rendering many of the receptors on the oocyte surface inactive. This is consistent with the total loss of currents in annelid and mollusk FaNaCs carrying mutations at various sites in the interfacial palm/β-ball site[21].

**Partial agonists bind via a similar mechanism to FMRFa**
Several other neuropeptides gate certain FaNaCs with lower potency and efficacy than FMRFa, including FVRIamides at annelid FaNaCs and FLRFa at mollusk FaNaCs[19,21,31,32]. We examined the structural basis of this partial agonism by solving the cryo-EM structure of FaNaC1 in the presence of ASSFVRIa, a product of the FVRIamide precursor in several annelids that gates FaNaC1 with relatively low potency and efficacy (Fig. 4a). The FaNaC1/ASSFVRIa cryo-EM map was resolved to 2.4 Å resolution, with a discrete density in the same ligand-binding pocket as that described for FMRFa (Fig. 4b and Extended Data Fig. 8). This density was best fit by the C-terminal FVRIa segment of the peptide, and the N-terminal ASS segment was not resolved (Fig. 4c,d and Extended Data Fig. 8). This indicates a very similar binding mechanism for both full and partial agonists, whereby FaNaC1 V122 and F129 coordinate the hydrophobic C-terminal side chain—F4 in FMRFa and I7 in ASSFVRIa

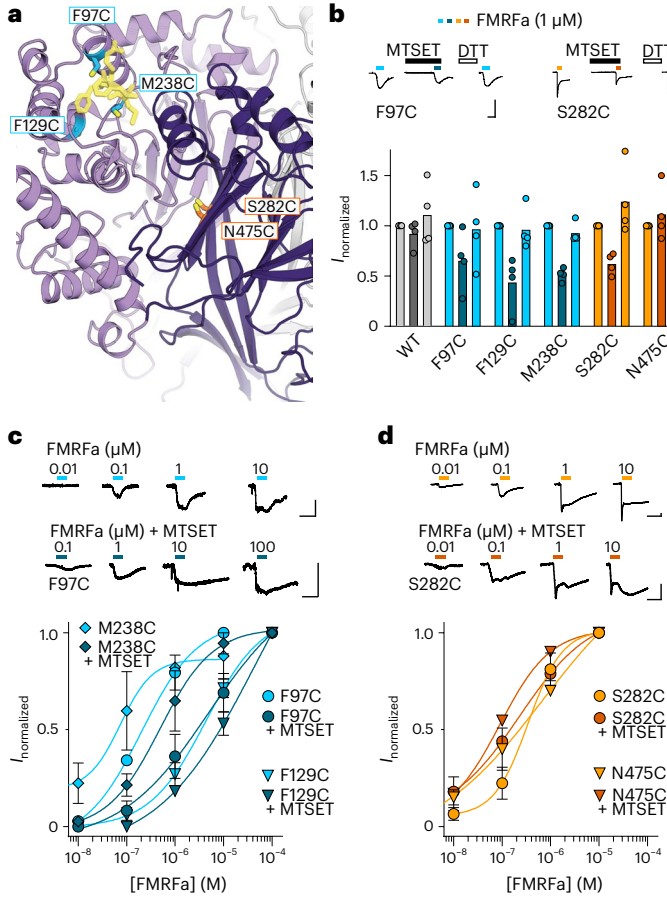

**Fig. 3 | Chemical modification of FMRFa binding site and intersubunit interface. a**, Illustration of cysteine substitutions in FMRFa binding site and previously identified interfacial site. **b**, FMRFa-gated currents with and without MTSET (300 μM) and after dithiothreitol (DTT) (2 mM) in oocytes expressing indicated FaNaC1 constructs. Top, example recordings. Bottom, mean (columns) and individual (dots, $n = 4$ oocytes) current amplitude normalized to that in the absence of MTSET. Scale bars, $x$, 10 s; $y$, 100 nA. **c,d**, Top, concentration-dependent FMRFa-gated currents in oocytes expressing indicated FaNaC1 constructs. Bottom, mean (±s.e.m., $n = 4$ oocytes) concentration-dependent current amplitude normalized to maximum current in the absence of MTSET. Scale bars in **c** and **d**, $x$, 10 s; $y$, 50 nA.

(Fig. 4c,d). We confirmed that the four C-terminal residues of ASSFVRIa determine its agonist activity by measuring FaNaC1 responses to FVRIa, observing very similar activity to the parent peptide (Fig. 4a). Similar binding mechanisms for neuropeptides with loosely conserved hydrophobic C-terminal residues and divergent N-terminal segments explains how annelid FaNaCs are gated by diverse neuropeptides, including FMRFa, various FVRIamides and LFRYa (ref. 21).

Full agonist FMRFa and partial agonist ASSFVRIa induce highly similar FaNaC1 conformations (Fig. 4c). In addition to the high similarity between FaNaC1/FMRFa and FaNaC1/ASSFVRIa structures, neither preparation yielded additional distinct 3D classes or any indication of conformational heterogeneity in the image processing (Extended Data Figs. 2 and 8). Thus, the partial agonism of ASSFVRIa does not appear to derive from the induction of a different conformational state compared to FMRFa.

### A putative open-channel state

To establish how ligand binding induces channel gating, we compared ligand-free and ligand-bound FaNaC1 structures. In both ligand-bound structures, however, the channel pore appears closed, with radii of ~1 Å at the level of G503 and G506 in TM2a (G3′ and G6′ in a TM2 numbering

scheme[33]; Fig. 5a). This is similar to our ligand-free, inactive FaNaC1 structure (Fig. 5a) and is too narrow to pass even mostly dehydrated Na⁺ ions (~2.3 Å). Thus, FMRFa- and ASSFVRIa-bound channels have probably adopted a desensitized state in the prolonged presence of agonist, in accord with the large decrease in current amplitude that occurs within ~2 s of FMRFa application, especially at higher FMRFa concentrations (for example, Figs. 1a, 2b, 4a and 5b). We think this rapid decrease in current is desensitization rather than channel block of FaNaC1 by the cationic moiety of FMRFa, as it also occurs at positive membrane potentials and also with noncationic derivatives of FMRFa (Extended Data Fig. 5c,d).

We therefore sought structural data on a ligand-bound, open-channel state by solving the structure of FaNaC1 in the presence of both FMRFa and diminazene, a pore blocker of diverse DEG/ENaC channels[20,34,35] that delays desensitization in ASICs by plugging the open channel pore[35]. We first verified that diminazene blocks FaNaC1 expressed in *Xenopus* oocytes by co-applying FMRFa and diminazene and observed inhibition of both peak (IC₅₀ = 9 ± 3 μM, $n = 5$) and sustained FMRFa-gated current (IC₅₀ = 1.8 ± 0.9 μM, $n = 6$; Fig. 5b and Extended Data Fig. 9a,b). Diminazene block was stronger at negative membrane potentials, and a large rebound current was observed after the removal of FMRFa and diminazene (Fig. 5b), suggesting that the positively charged drug inhibits FaNaC1 by plugging the open-channel pore and preventing the channel closure that occurs in desensitization.

In solving the FaNaC1/FMRFa/diminazene structure, several 3D classes emerged during our image analysis, with two predominating: a closed-channel class similar to the FaNaC1/FMRFa structure; and a class that differed from the others with dilated mid- to upper pore, and some probably nonprotein density in the channel pore (Fig. 5c and Extended Data Fig. 10). We focused on this second 3D class, resolving the structure to 3 Å resolution (Extended Data Fig. 10). We easily modeled TM2 helices into the cryo-EM density and observed an ~1 Å increase in the pore radius relative to our other structures (Fig. 5a). Although the resolved density in the pore was too small to accommodate a full diminazine molecule (Fig. 5b,c), no such density or dilated conformation was observed in the diminazine-free FMRFa-bound dataset (Extended Data Fig. 2), suggesting that diminazene binds and affects FaNaC1 pore conformation.

Given the drug's effect on pore radius and its voltage-dependent block of currents, we hypothesized that this nonprotein density derives from partially unresolved or low occupancy diminazene molecules. To investigate this further, we measured diminazene block of mutant FaNaC1 channels and observed that increasing side chain volume around the density via the G6′S substitution drastically reduced, and via the G3′S substitution one helical turn higher modestly reduced, diminazene potency (Fig. 5c,d and Extended Data Fig. 9c). This suggests that diminazene binds intimately at the level of G6′. We also generated mutant G2′S channels but saw no currents in oocytes injected with these RNAs ($n = 8$ over two batches of ooyctes). An additional helical turn higher, the S-1′A substitution caused constitutive current, as expected for mutating the TM2a-1′ *degenerin* position[36], but had no effect on diminazene potency (Fig. 5d and Extended Data Fig. 9c). In contrast, the D0′N substitution decreased diminazene potency approximately tenfold (Fig. 5d), suggesting that the relatively well-conserved TM2a 0′ carboxylate is important for sensitivity to diminazene. This may explain why ENaCs, which instead possess asparagine at the 0′ position, are insensitive to diminazene[37] and closely reflects computational docking of the drug to ASIC1, where the D0′ carboxylate engages the upper positively charged amidine moiety[35]. Thus, we interpret our FaNaC1/FMRFa/diminazene structure as an FMRFa-gated, diminazene-blocked, open-channel conformation.

Based on the pore radius of ~2 Å at G3′, G6′ and the re-entrant loop in our putatively open-channel structure (Fig. 5a), FaNaC1 presumably passes partly dehydrated Na⁺ ions and its pore is narrower than the previously captured open-channel structure of ASIC1 (ref. 11),

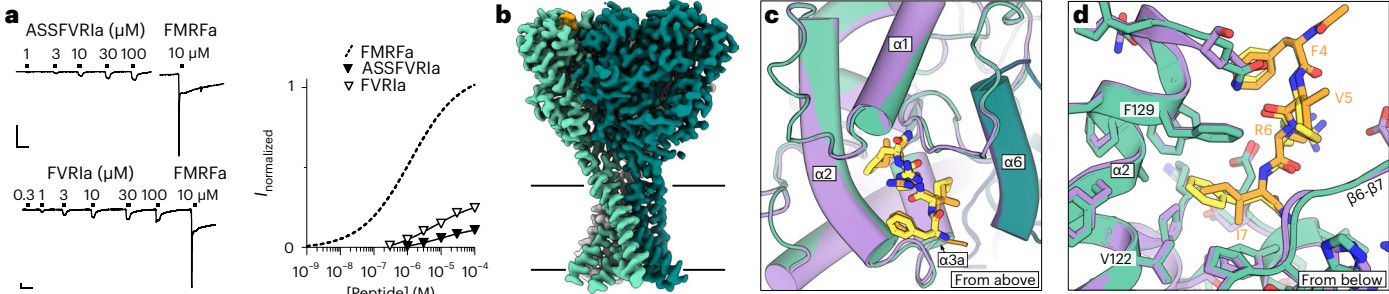

**Fig. 4 | Partial agonist binding in the FMRFa site. a**, Left, FaNaC1 currents gated by ASSFVRIa, FVRIa and FMRFa. Scale bars, *x*, 30 s; *y*, 250 nA. Right, average (± s.e.m., *n* = 4 oocytes for FVRI and 5 oocyes for ASSFVRIa) ASSFVRIa- and FVRI-gated current amplitude normalized to 10 µM FMRFa on the same cells. FMRFa

curve repeated from Fig. 1a for display. **b**, Cryo-EM map of FaNaC1/ASSFVRIa. FaNaC1 subunits shaded in greens, ASSFVRIa orange. **c**, Overlay of FaNaC1/FMRFa (purple/yellow) and FaNaC1/ASSFVRIa (teal/orange) models viewed from above. **d**, Magnified view of overlay from **c** viewed from below.

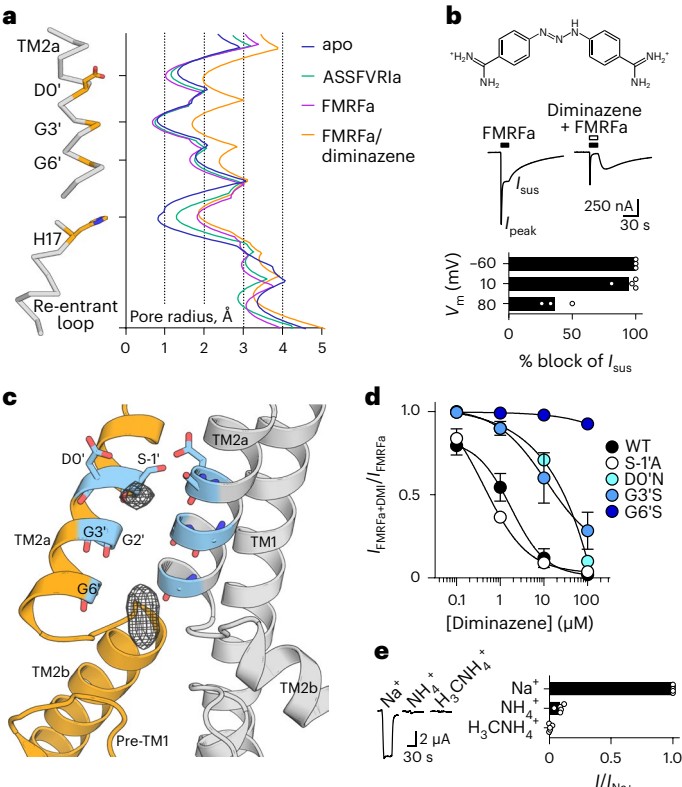

**Fig. 5 | FaNaC1/FMRFa/diminazene structure. a**, Pore-lining residues in FaNaC1/FMRFa/diminazene structure (left) and pore radius for all four cryo-EM structures calculated in HOLE[53] (right). **b**, Top, diminazene structure (ionized at pH 7.4). Middle, peak ($I_{peak}$) and sustained ($I_{sus}$) FMRFa (3 µM)-gated currents through FaNaC1 in the absence and presence of diminazene (10 µM). Bottom, % block (individual data points, *n* = 3 oocytes at −60 mV, 4 at 10 mV, 3 at 80 mV; and means, bars) of $I_{sus}$ by diminazene at three different membrane potentials. **c**, Magnified view of FaNaC1/FMRFa/diminazene model, showing nonprotein cryo-EM density in the pore as mesh. **d**, Concentration-dependent block of $I_{sus}$ by diminazene at WT and mutant channels (mean ± s.e.m., *n* = 4 oocytes, except D0′N—see Source data). **e**, FMRFa-gated current amplitude ($V_m$ = −60 mV; individual data points (*n* = 5) and means, bars) in extracellular solutions based on different cations.

although the re-entrant loop was not resolved in the latter. Furthermore, we found that larger nitrogen-based cations, ammonium and methylammonium, were much less permeant than Na⁺ in FaNaC1 (Fig. 5e), as observed for ENaC[38] but different from ASIC1, which

conducts substantial ammonium and methylammonium current[39,40]. This suggests that, compared to ASIC1, FaNaC1 adopts a narrower open-channel pore, as represented by our FMRFa-gated, diminazene-blocked structure, and that FaNaC1 presumably passes partly dehydrated Na⁺ ions.

### Ligand-induced channel gating

Finally, we compared ligand-free FaNaC1 and FaNaC1/FMRFa/diminazene structures to establish the structural mechanism by which FMRFa binding opens the channel. In its binding pocket, FMRFa draws the C-terminal end of α2 and α3a (distal finger domain) 3–5 Å upward and inward in the direction of α6 of the adjacent subunit, whereas the N-terminal end of α2, α3b/α3c and all of α6 (knuckle) are relatively static (Fig. 6a,b). This in turn draws the peripheral and long, vertical α5 segment (thumb) ~3 Å inward and ~2 Å upward (Fig. 6a,b). In contrast to the peripheral thumb, the more internal loops and β-sheets of the palm and β-ball domains remain relatively static, exemplified by F74 and Y174 side chains in a hydrophobic hub within this region (Fig. 6b). The net result of peripheral movement on internal stasis is that the extracellular domain of a single subunit rolls anticlockwise (viewed from above) and upward (viewed from the side, Fig. 6c). Extracellular domain rolling pulls the β-turn at the proximal-thumb/wrist domain outward, which couples to upper-channel expansion via β-turn H297 interactions with TM1 Y59 and TM2a E490 (Fig. 6b,c).

Seeking verification that β-turn–TM1/TM2 interactions mediate channel gating, we tested the activity of mutant H297S channels in which the large H297 side chain is replaced with a much smaller side chain, making these interactions less likely. H297S channels were constitutively active, with 3.8 ± 1.3 µA (*n* = 10) current in the absence of agonist (120 ± 40 nA in WT, *n* = 10), which was blocked on average 76 ± 5% (*n* = 7) by 30 µM diminazene (Fig. 6d). Furthermore, the addition of FMRFa activated additional current, with increased potency compared to WT, and the relative efficacy of partial agonist ASSFVRIa was increased by the mutation (Fig. 6d). Thus, the H297S mutation increases gating efficacy. This suggests that WT resting FaNaC1 channels are energetically primed for opening but cannot do so until ligand binding and extracellular domain rolling releases TM1/TM2a via outward H297 movement.

We find that ligand-induced channel gating extends to the lower part of the pore, observing a 1 Å increase in pore radius at the level of re-entrant loop T15 and H17 side chains compared to the ligand-free state (Fig. 5a). This offers structural evidence for lower-pore gating that was suggested to occur in molluscan FaNaC and mammalian ENaC based on electrophysiological studies[41–43]. Lower-pore dilation is similar in both FaNaC1/FMRFa/diminazene and FaNaC1/FMRFa structures (Fig. 5a), in contrast to upper-pore dilation, which is present in the FaNaC1/FMRFa/diminazene structure but collapsed in the desensitized

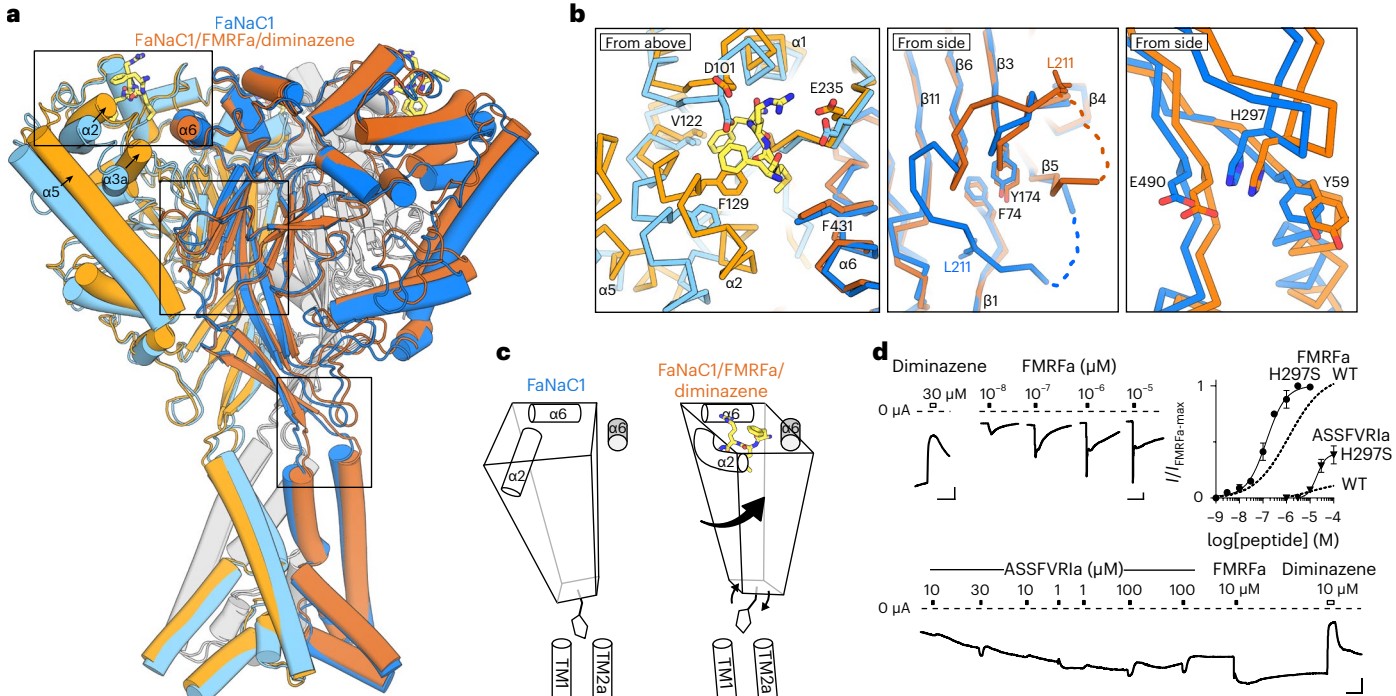

**Fig. 6 | FaNaC1 gating mechanism. a**, Overlay of ligand-free FaNaC1 (blue) and FaNaC1/FMRFa/diminazene (orange) structures. Adjacent subunit in darker shades. **b**, Magnified views of boxed regions, left to right: ligand binding site; palm domain; β-turn–TM2a/TM1 interactions. Selected side chains shown as sticks and labeled. **c**, Illustration of one subunit (left subunit in **a**) moving during gating: peripheral rolling, internal stasis and β-turn moving outward to allow channel expansion. α6 helix from adjacent subunit in gray. **d**, Top left and bottom, example recordings from oocytes expressing mutant H297S FaNaC1 channels (scale bars, *x*, 30 s; *y*, 1 μA). Top right, mean (± s.e.m., *n* = 5 oocytes) responses of H297S channels to FMRFa and ASSFVRIa normalized to maximum FMRFa-gated current ($I/I_{max-FMRFa}$), compared to WT channels. WT curves repeated from Figs. 1a and 4a.

FaNaC1/FMRFa structure (Fig. 5a). This suggests that desensitization is an upper-channel and extracellular vestibule phenomenon. We also observed a remarkably large conformational change in the β5–β6 loop (residues K200–G218) of the extracellular domain, which flips from a downward orientation in our ligand-free structure to an upward orientation in all ligand-bound structures (Fig. 6a,b). This involves, for example, L211 displacing 20 Å (Fig. 6b) and is a notable exception to the otherwise immobile palm/ball domain. Whether β5–β6 loop flipping is a consequence of activation or a diminazine-insensitive aspect of desensitization would require further investigation.

## Discussion

### Structural basis for diverse activators of DEG/ENaCs

Our four *Malacoceros* FaNaC1 structures offer a precise description of neuropeptide binding and a comprehensive view of ligand-induced channel gating in a DEG/ENaC channel. The extracellular neuropeptide-binding pocket is formed by dynamic α1–α3 segments of the finger domain and more static β6–β7 and α6 segments of proximal-finger and knuckle domains. Comparison with other DEG/ENaCs shows similar overall channel architecture but divergent sequence and secondary and tertiary structure in the first three helices of the finger domain (Extended Data Fig. 6). Despite its divergence in different DEG/ENaCs, this external corner of the DEG/ENaC trimer appears to play important roles in gating throughout the superfamily. It includes modulatory ion- and protease-binding sites in ENaC[15,44], residues whose mutation decreases pH sensitivity in ASICs[45], and proposed sites for extracellular matrix tethering to mechano-sensitive DEG/ENaCs[46].

α1–α3 amino acid sequence is even divergent between closely related annelid FaNaCs and mollusk FaNaCs (Extended Data Fig. 7). Despite this divergence, α1–α3 helices form an architecturally very similar FMRFa-binding pocket in both *Malacoceros* (annelid) FaNaC1 and *Aplysia* (mollusk) FaNaC[30]. But whereas FMRFa F4 orients most deeply into the site in *Malacoceros* FaNaC1, F1 orients most deeply in *Aplysia* FaNaC. This is reflected in previous work showing that annelid FaNaCs are gated by partial agonists containing N-terminal additions such as PSSFVRIa and LFRYa[21], in contrast to mollusk FaNaCs, which are instead gated by partial agonists more closely analogous to FMRFa[19,21,31,32]. Because of this divergence within α1–α3 helices, binding site residues whose mutations decrease FMRFa potency, such as *Malacoceros* FaNaC1 α2-V122 and α2-F129 and several α1–3 residues in mollusk FaNaCs[30,47–49], actually occupy different orientations in space or are even absent from various cousins within the broader FaNaC family (Extended Data Fig. 7). Perhaps surprisingly, the more strictly conserved determinants of agonist potency throughout the FaNaC family, whose mutation decreases FMRFa activity in various FaNaCs, are in a different site, between palm and β-ball domains of adjacent subunits[21]. Our combined structural and experimental study now shows these conserved residues are probably important for coupling ligand binding in the finger domain to channel gating further below and not for ligand binding.

### Channel gating and ion conduction illuminated by FaNaC1

Previously, our understanding of DEG/ENaC gating mechanisms and ion conduction was based on chicken ASIC1 structures in resting, active and desensitized states[11,12,25]. The FaNaC1 structures presented here capture a ligand-free resting state, two ligand-bound desensitized states and a ligand-bound dilated channel state. The comparison of these structures points to a gating mechanism in which the outer finger domain closes around the agonist, starting an anticlockwise rotation of the extracellular domain (Fig. 6). As the palm remains static, the periphery of the extracellular domain (the thumb) rolls anticlockwise and upward

with the finger helices, pulling the upper part of the channel domain (wrist) outward, resulting in pore dilation (Fig. 6c). This anticlockwise extracellular domain rotation and wrist expansion loosely reflects gating of ASIC1 (ref. 12), although the principal trigger(s) initiating these conformational changes in ASIC1 are yet to be identified[13].

Despite macroscopic desensitization of FaNaC1 reflecting that of several ASICs, with fast entry to desensitization and then a noticeable sustained current[4], the structural basis for desensitization may differ between FaNaCs and ASICs. The large rearrangement of the β11–β12 linker during ASIC1 desensitization[12] is not observed in FaNaC1, although we do observe a large conformational change of the FaNaC1 β5–β6 loop, which is close to the β11–β12 linker (Fig. 6b). We notice concentration-dependent desensitization in FaNaC1, with little and large decreases in current amplitude in low and high FMRFa concentrations, respectively, and with extremely slow washout of currents after high concentrations of FMRFa (Extended Data Fig. 5). Rapid current decay and smaller current amplitudes with high ligand concentrations in FaNaC1 did not seem related to pore block by the ligand, in contrast to what has been suggested for mollusk FaNaCs[26,30,48].

An advance of our study is the capture of a potentially open-channel pore structure. In previous high-resolution DEG/ENaC structures the channel domain was difficult to resolve[11,12,15], and in particular, the only open-channel structure of chicken ASIC1 does not include the pore-lining pre-TM1 re-entrant loop[11]. Nonetheless, a better resolved picture of ion conduction throughout the superfamily emerges from the comparison of FaNaC1 and distantly related ASIC1 pores. FaNaC1 is more closely related to ENaC[7,34,50], both of these channels are poorly permeable to nitrogen-based cations larger than Na+ (ref. 38), and our FaNaC1/FMRFa/diminazene structure reveals an ion pathway slightly narrower than both ASIC1 and voltage-gated sodium channels[11,51], both of which pass ammonium, methylammonium and hydroxylamine relatively well[39,40,52]. Taken together, our capture of FaNaC1 in different functional states describes the mechanism by which FMRFa elicits excitatory neuronal signals and offers a template for future studies dissecting channel gating and ion conduction in DEG/ENaCs.

## Online content

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

## Methods

### Cell lines

Adherent HEK293T cells (CRL-3216, American Tissue Culture Collection) were cultured in 10-cm Petri dishes in Dulbecco's modified Eagle medium with L-glutamine and sodium pyruvate (Gibco), supplemented with 10% fetal bovine serum (FBS) and antibiotic–antimycotic at 37 °C and 5% $CO_2$. Suspension HEK293S GnTI⁻ cells (CRL-3022, American Tissue Culture Collection) were maintained in Freestyle medium with GlutaMAX (Gibco) supplemented with 1% FBS and antibiotic–antimycotic solution, at 37 °C, 5% $CO_2$ and 60% humidity, in TPP600 bioreactors. Sf9 cells (12659017, ThermoFisher Scientific) were cultured in SFMIII medium supplemented with antibiotic–antimycotic, at 27 °C.

### Protein expression and purification

Commercially synthesized *Malacoceros fuliginosus* FaNaC1 coding sequence (Supplementary Notes) was cloned into a pEZT-BM vector[54] adapted for FX cloning[55] with C-terminal HRV-3C cleavage site, Venus YFP, myc and SBP tags. For expression screening, adherent HEK293T cells at ~60% confluency were transfected with this plasmid using PEI 40K MAX (DNA:PEI ratio of 1:3, 10 µg DNA per dish). Protein was expressed for 48 h, cells were collected, washed with phosphate-buffered saline and stored at −80 °C until further use. Cells from one dish were resuspended in 200 µl extraction buffer (2% *n*-dodecyl β-maltoside (DDM), 0.4% cholesteryl hemisuccinate (CHS), 20 mM HEPES pH 7.6, 150 mM NaCl, 10% glycerol and cOmplete protease inhibitors), proteins were extracted for 2 h. Lysate was centrifuged at 150,000$g$ and supernatant was analyzed on Tosoh G4000PWXL using fluorescence size-exclusion chromatography[56].

Large-scale expression of *Malacoceros* FaNaC1 was performed using BacMam expression system[57]. Virus was generated as described[54,57]. Briefly, the *Malacoceros* FaNaC1 bacmid was generated following the Invitrogen Bac-to-Bac protocol. Afterward, Sf9 cells were transfected with the bacmid using Cellfectin according to the manufacturer's instructions. Four to five days after transfection, the P0-containing supernatant was collected and supplemented with 10% FBS, and this virus stock was used to generate P1. Sf9 cells were infected at density $1 \times 10^6$, and once the majority of cells were fluorescent, the supernatant containing virus was filtered, supplemented with 10% FBS and stored at 4 °C until further use. One day before infection, HEK293S were split to $0.6 \times 10^6$ cells per ml density. The following day, titerless P1 virus was diluted 1:10 into expression culture. After 24 h, sodium butyrate was added to a final concentration of 10 mM and the protein was expressed for additional 48 h. Cells were collected, washed with phosphate-buffered saline and stored at −80 °C.

All of the purification steps were performed on ice or at 4 °C. Cell pellets from ~6 l of expression culture were resuspended in buffer A (20 mM HEPES pH 7.6, 150 mM NaCl, 10% glycerol, DNase I, 2 mM $MgCl_2$, 2% DDM, 0.4% CHS and cOmplete protease inhibitor tablets), and the protein was extracted for 2 h under gentle agitation. The lysate was centrifugated at 200,000$g$ for 30 min, the supernatant was applied to the 3K1K resin (GFP enhancer nanobody produced in *Escherichia coli* as described[58] and coupled to NHS-sepharose (Cytiva) according to manufacturer's instructions). The supernatant was incubated with the resin for 30 min and passed through the resin three to four times in a gravity flow column. The resin was washed with ~30 column volumes of buffer B (20 mM HEPES pH 7.6, 150 mM NaCl, 10% glycerol and 0.02% glyco-diosgenin (GDN)). Afterward, the protein was cleaved off the resin in batch with HRV-3C protease (~1.2 mg) for 2 h. The eluate was concentrated using 100 kDa cutoff Amicon centrifugal filter units at 600$g$ and injected onto a Superose 6 Increase 10/300 column equilibrated in buffer C (20 mM HEPES pH 7.6, 150 mM NaCl and 0.02% GDN). Main peak fractions were pooled and concentrated as described above.

### Nanodisc reconstitution

Nanodisc reconstitution was performed as described[59]. Briefly, lipids (POPC:POPG 3:1 molar ratio) were pooled, dried using rotary evaporator, and washed with diethyl ether. After diethyl ether was evaporated, the lipids were rehydrated in ND buffer (20 mM HEPES, 150 mM NaCl and 30 mM DDM) at a concentration of 10 mM. Purified protein was mixed with lipids and incubated for 30 min, after which the purified MSP was added and incubated for 30 min, followed by addition of SM-2 biobeads (200 mg ml⁻¹ of assembly reaction). The assembly ratios FaNaC:lipids:MSP were 3:1,100:10, assuming 1 FaNaC trimer per five assembled nanodiscs. The mixture was incubated overnight at 4 °C with gentle agitation. The following day, the sample was concentrated in 100 kDa Amicon concentrators at 500$g$ and injected onto a Superose 6 Increase 10/300 column equilibrated in buffer D (20 mM HEPES pH 7.6 and 150 mM NaCl). Higher molecular weight peak containing nanodisc-reconstituted FaNaC1 was pooled and concentrated as above to 1.4–1.8 mg ml⁻¹. Ligands and diminazene were added directly before freezing (FMRFa 30 µM, ASSFVRIa 100 µM and diminazene 100 µM).

### Cryo-EM sample preparation and data acquisition

Quantifoil 1.2/1.3 Au grids with 300 mesh were glow-discharged at 5 mA for 30 s. The grids were prepared using a Vitrobot Mark IV (Thermo Fisher). For that, 2.8 µl of freshly prepared sample was applied to grids, which were blotted for 3.5 s with a blot force 0 at 15 °C and 100% humidity. The grids were plunge-frozen in an ethane–propane mixture and stored in liquid nitrogen until further use. The data were recorded at the University of Groningen on a 200 keV Talos Arctica (Thermo Fisher) with a K2 summit direct detector (Gatan), a post-column energy filter with a 20 eV slit and a 100 µm objective aperture. Optimal squares and holes for data collection were selected using an in-house sample thickness estimation script[60]. The images were recorded in an automated fashion using SerialEM v.3.9.0 beta[61] with a 3 × 3 multi-shot pattern. Cryo-EM images were acquired at a pixel size of 1.022 Å (calibrated magnification of ×49,407), a defocus range from −0.5 to −2 µm, an exposure time of 9 s with a subframe exposure time of 150 ms (60 frames) and a total electron exposure on the specimen of about 52 electrons Å⁻². Micrographs were preprocessed on the fly in FOCUS v.1.1.0 (ref. [62]) using MotionCor2 v.1.4.0 (ref. [63]) for motion correction and ctffind4.1.14 (ref. [64]) for contrast transfer function (CTF) resolution estimation. Images with defocus values 0.4–2 µm, showing no ice contamination and a CTF resolution estimate better than 6 Å were selected for further processing.

### Image processing

The collected datasets were processed following an essentially identical scheme, with the exception of the FaNaC1/FMRFa/diminazene dataset. Particles were picked using a general model in crYOLO v.1.8.2 (ref. [65]), and subsequently extracted in Relion v.3.1.0 (ref. [66]) with a box size of 220 pixels for FMRFa, ASSFVRIa and FMRFa/diminazene datasets, and 240 pixels for apo, respectively. The extracted particles were imported into cryoSPARC v.3 (ref. [67]) and subjected to 2D classification (initial batch size 200, ten final full iterations). Particles from selected 2D classes were subjected to ab initio 3D reconstruction with five classes and subsequent heterogeneous refinement using all five ab initio classes as an input. Particles from the best class were imported into Relion v.3.1, and subjected to Bayesian polishing followed by several rounds of CTF refinement. In the case of the diminazene-supplemented dataset, particles were further classified with no image alignment and a mask covering the transmembrane part to resolve the conformation heterogeneity in the pore region. In all cases, the final sets of particles were subjected to masked refinement with a C3 symmetry imposed. The half-maps were used as inputs for postprocessing in deepEMhancer 220530_cu10 (ref. [68]) with a tight model. Resolution was estimated in Relion v.3.1 postprocessing, using a mask excluding nanodisc density

according to the standard Fourier shell correlation cut-off of 0.143 (Extended Data Figs. 1, 2, 8 and 10).

## Model building and refinement and pore analysis

The initial model of *Malacoceros* FaNaC1 was predicted using AlphaFold 2 (ref. 69) and adjusted manually in Coot v.0.9.8.1 (ref. 70). Models were iteratively adjusted in Coot and ISOLDE v.1.6.0 (ref. 71), followed by real-space refinement in Phenix v.1.20.1-4487 (ref. 72) with non-crystallographic symmetry and secondary structure restraints against a refined unsharpened map. Figures were prepared in Pymol v.2.5.5, Chimera v.1.16 (ref. 73) and ChimeraX v.1.5 (ref. 74). Channel pore radius was calculated with HOLE[53] implemented in Coot.

## Electrophysiology and data analysis

FaNaC1 in a modified pSP64 plasmid vector (Supplementary Notes) was used for mutagenesis and messenger RNA preparation and injection into stage V/VI *Xenopus laevis* oocytes (EcoCyte Bioscience). Mutants were generated by partly overlapping primers (Supplementary Table 1 and ref. 75), and channel-coding inserts were Sanger sequenced. FMRF–NH$_2$, ASSFVRI–NH$_2$, FVRI–NH$_2$, AMRF–NH$_2$, FMRA–NH$_2$, FMQF–NH$_2$ and FM(citrulline)F–NH$_2$ (acetate salts, custom synthesized by Genscript, purity 95.1–99.6% purity by high-performance liquid chromatography, mass confirmed by electrospray ionization–mass spectrometry) were dissolved in water to 10 mM and diminazene aceturate (Merck) was dissolved in dimethyl sulfoxide to 100 mM before dilution in experimental solution: (in mM) NaCl 96, KCl 2, CaCl$_2$ 1.8, MgCl$_2$ 1 and HEPES 5, pH 7.5 (NaOH). NaCl was replaced with KCl, NH$_4$Cl or H$_3$CNH$_3$Cl (Merck) where appropriate.

Oocytes were clamped at −60 mV unless otherwise indicated, and currents were measured by two-electrode voltage clamp, as previously described[21], using a Warner OC-725C amplifier and HEKA LIH8 + 8 interface with Patchmaster 2x90.4 software (HEKA), sampling at 500 Hz or 1 kHz and filtering at 100 Hz. Current amplitudes were measured in Clampfit v.11 (Molecular Devices). *N* values refer to experiments performed on different oocytes. Data were analyzed in Prism v.9 (GraphPad) and fit with Prism v.9 variable-slope four-parameter nonlinear regression, yielding half-maximal effective activating/inhibiting concentrations (EC$_{50}$/IC$_{50}$). In establishing EC$_{50}$ values for activation by FMRFa, we waited 4 min between applications of increasing concentrations. As detailed in Extended Data Fig. 5a,b, this risked incomplete recovery from desensitization at high ligand concentrations, but it ensured better recordings without substantially affecting EC$_{50}$. Mean ± standard error of the mean (s.e.m.) EC$_{50}$/IC$_{50}$ from fits to individual cells reported in Main. Fits to averaged data points shown in figures. Currents shown in figures are further filtered (20 Hz) and decimated (50×) in Clampfit 11 for smaller file size.

## Reporting summary

Further information on research design is available in the Nature Portfolio Reporting Summary linked to this article.

## Data availability

The *Malacoceros fuliginosus* FaNaC1 coding sequence is available in GenBank (entry ON156825.1) and shown in Supplementary Notes. Ligand-free, FMRFa-bound, ASSFVRIa-bound and FMRFa-bound+diminazene structures are available in PDB via entries 8ON8, 8ON7, 8ON9 and 8ONA, and in EMDB via entries 16982, 16981, 16983 and 16984, respectively. Micrographs were deposited to EMPIAR under the following accession codes: 11631 (apo FaNaC1), 11632 (FMRFa-bound FaNaC1), 11633 (ASSFVRIa-bound FaNaC1) and 11634 (FMRFa-bound FaNaC1 in the presence of diminazene). Source data are provided with this paper.

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

## Acknowledgements

V.K. was supported by SNSF Postdoc.Mobility Fellowship no. P500PB_203053, C.P. by the Dutch Research Council (NWO) grant nos. 722.017.001 and 740.018.016, and T.L. by The Research Council of Norway, project no. 234817. We thank J. Rheinberger and M. Punter for facility and image processing cluster management. Members of the Paulino and Lynagh laboratories are acknowledged for technical help and discussions at various stages of the project.

## Author contributions

V.K. was responsible for project conception and design, molecular biology, protein expression and purification, cryo-EM data collection and analysis, figure preparation and manuscript editing. M.D. was responsible for electrophysiology experiments and analysis, figure preparation and manuscript editing. C.P. was responsible for supervision and manuscript editing. T.L. was responsible for project design, supervision, molecular biology, electrophysiology experiments and analysis, figure preparation and manuscript writing.

## Competing interests

The authors declare no competing interests.

## Additional information

**Extended data** is available for this paper at https://doi.org/10.1038/s41594-023-01198-y.

**Correspondence and requests for materials** should be addressed to Cristina Paulino or Timothy Lynagh.

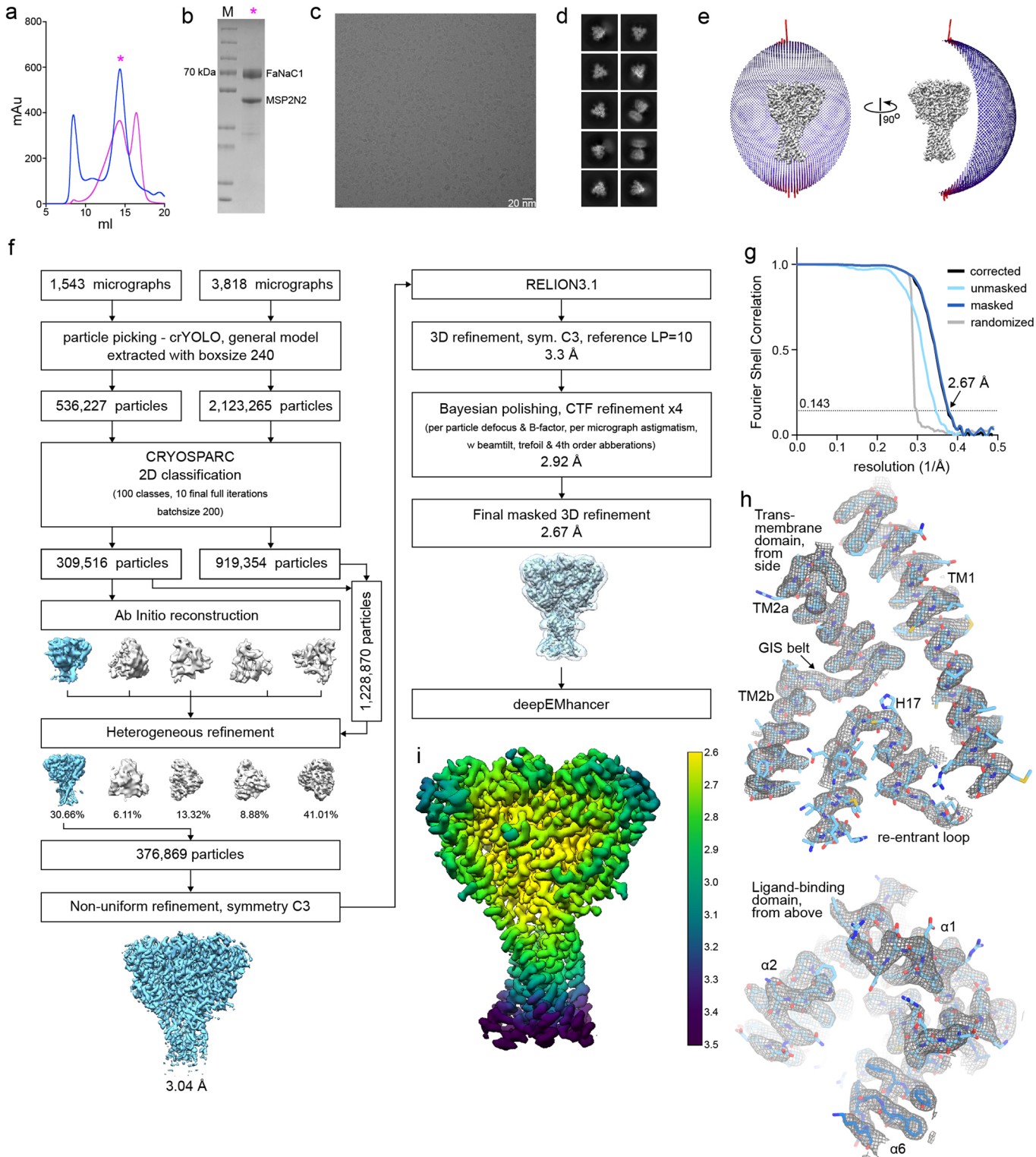

**Extended Data Fig. 1 | Ligand-free FaNaC1 structure determination by cryo-EM.** (**a**) Size exclusion profiles of detergent-solubilised (blue) and nanodisc-reconstituted (magenta) *Malacoceros* FaNaC1. Samples were analysed on a Superose 6 Increase 10/300 column. (**b**) SDS-PAGE of the main peak fraction, which was used for sample preparation for cryo-EM. Protein was purified and reconstituted into lipid nanodiscs three times with similar size exclusion profiles (a) and gel migration (b). Representative cryo-EM image of 5361 micrographs (**c**) and 2D classes (**d**) of vitrified FaNaC1 in apo state. (**e**) Angular distribution of the particles included in the final C3-symmetrised map. The length and the colour of the sticks represent the number of particles. (**f**) Schematic representation of the processing workflow, the mask used in the final refinement iteration is displayed as a transparent outline. (**g**) FSC plot used for resolution estimation (0.143 cut-off criteria). (**h**) Density corresponding to the transmembrane domain and ligand-binding domain shown as grey mesh, deepEMhancer map used for visualization is contoured at 4.5 σ. The respective fitted model is shown in blue. (**i**) Final deepEMhancer-postprocessed map coloured according to the local resolution estimation in Relion.

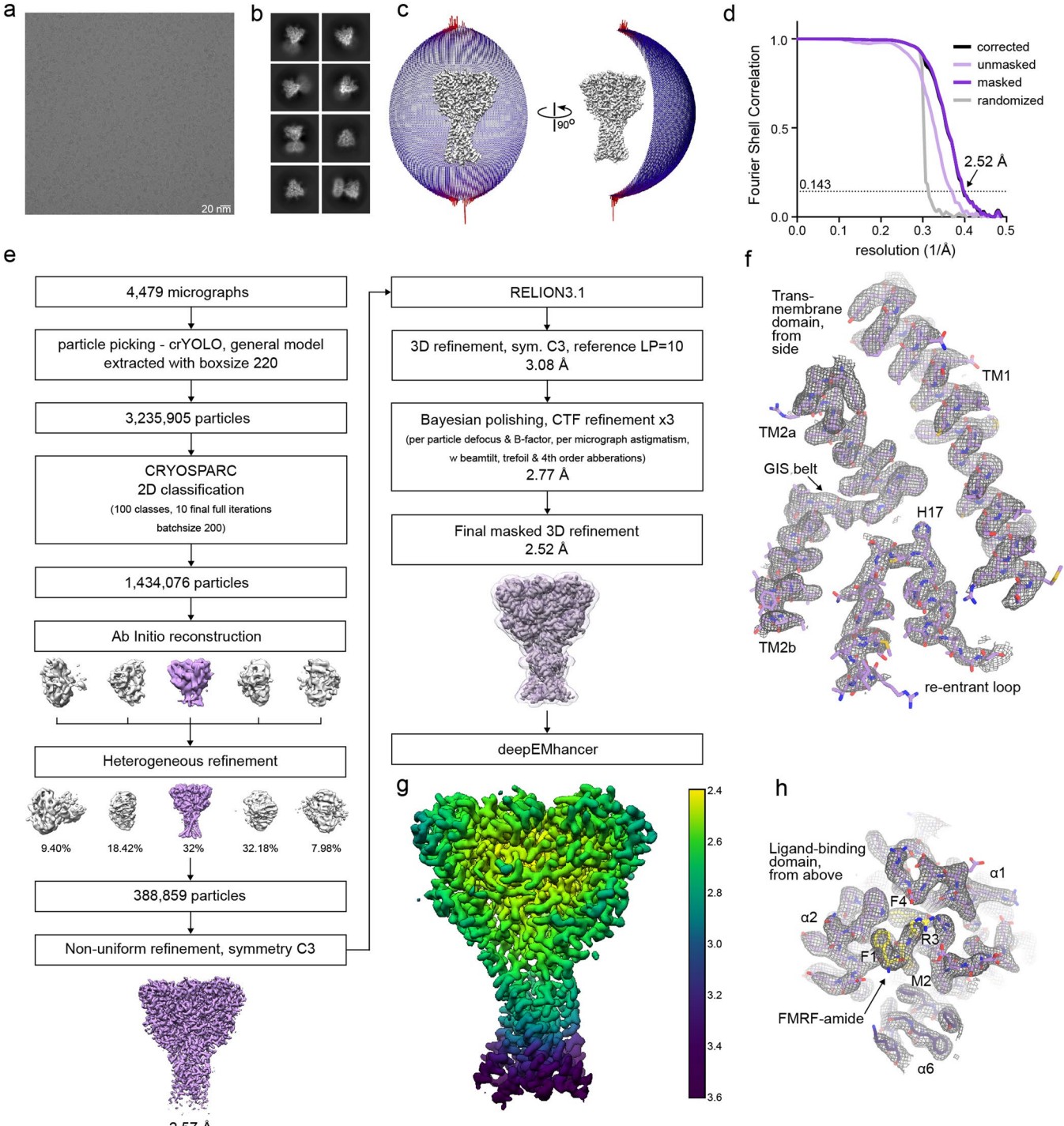

**Extended Data Fig. 2 | FMRFa-bound FaNaC1 structure determination by cryo-EM.** Representative cryo-EM image of 4479 micrographs (**a**) and 2D classes (**b**) of vitrified FaNaC1 in presence of FMRFa. (**c**) Angular distribution of the particles included in the final C3-symmetrised map. The length and the colour of the sticks represent the number of particles. (**d**) FSC plot used for resolution estimation (0.143 cut-off criteria). (**e**) Schematic representation of the processing workflow, the mask used in the final refinement iteration is displayed as a transparent outline. (**f**) Density corresponding to the transmembrane domain shown as grey mesh, deepEMhancer map used for visualization is contoured at 4.5 σ. The respective fitted model is shown in purple. (**g**) Final deepEMhancer-postprocessed map coloured according to the local resolution estimation in Relion. (**h**) Close-up view of the ligand-binding site, deepEMhancer map is displayed as grey mesh contoured at 4.5 σ, fitted atomic model is displayed in purple, FMRFa molecule is shown in yellow.

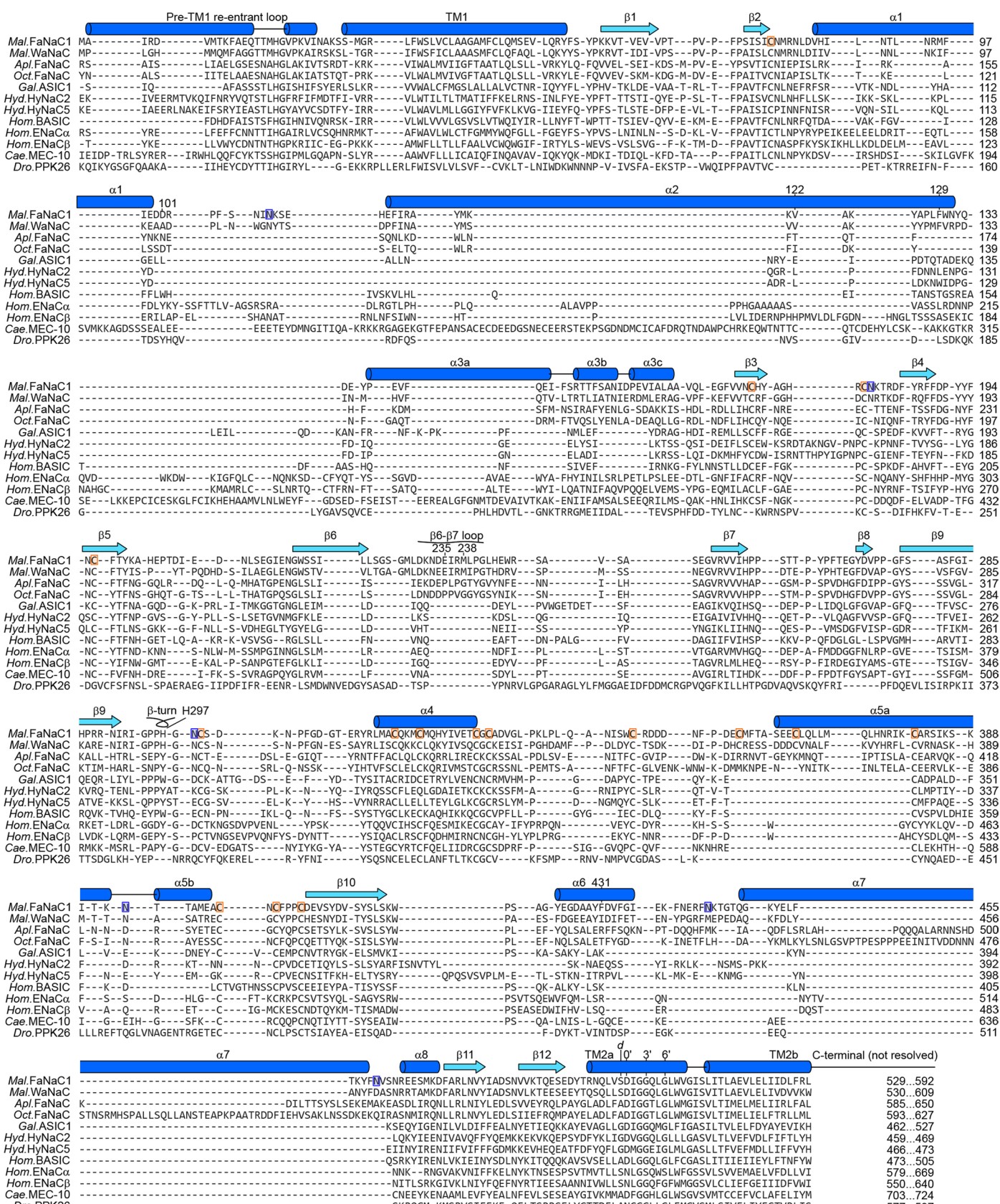

**Extended Data Fig. 3 | Amino acid sequence alignment of FaNaCs and other DEG/ENaCs.** *Malacoceros fuliginosus* (*Mal.*, bristle worm) FaNaC1 and WaNaC, *Aplysia kurodai* (*Apl.*, sea slug) FaNaC, *Octopus bimaculoides* (*Oct.*) FaNaC, *Gallus gallus* (*Gal.*, chicken) ASIC1, *Hydra vulgaris* (*Hyd.*) HyNaC, human (*Hom.*) BASIC and ENaC, *Caenorhabditis elegans* (*Cae.*) MEC-10, and *Drosophila melanogaster* (*Dro.*) PPK26 were extracted from an existing alignment of 544 DEG/ENaCs[21] and

manually adjusted in α1-α3 to reflect secondary structure (dark blue α-helices, light blue β-strands) of FaNaC1 (present work) and ASIC1[14]. Selected FaNaC1 residues discussed in main text are numbered. 'Prime' numbering (for example 0′, 3′, 6′) for better comparison of TM2 residues[33]. *d*, position of gain-of-function *degenerin* mutations[46]. Orange boxes, FaNaC1 disulfide-forming cysteines. Blue boxes, FaNaC1 glycosylation sites.

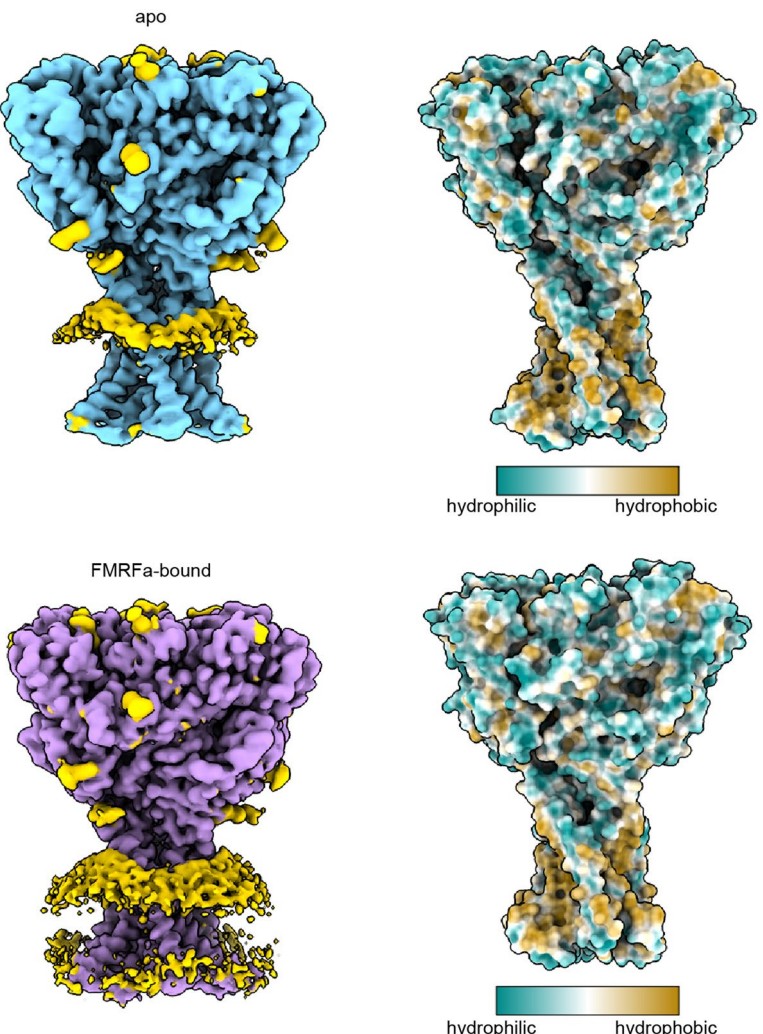

**Extended Data Fig. 4 | Effect of FaNaC1 on the surrounding lipid bilayer.** Shown are unmasked refined maps of FaNaC1 in apo- (blue) and FMRFa-bound (purple) states, density corresponding to nanodisc and glycosylation sites are shown in yellow. Right, surface representation of the refined models coloured according to hydrophobicity.

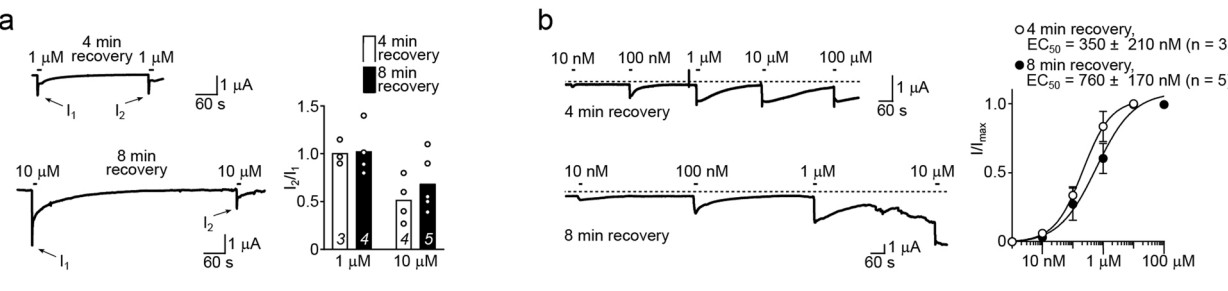

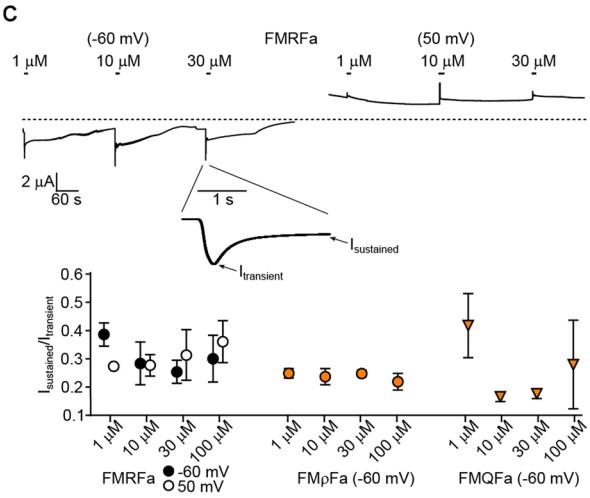

**Extended Data Fig. 5 | Measurement of FMRFa and other peptide potency.**
(**a**) *Left*, example recordings, and *right*, individual (points) and mean (columns; n = x oocytes indicated in italics on columns) amplitude of a second FMRFa-gated current ('$I_2$') relative to a first ('$I_1$') after four minutes or eight minutes recovery, for 1 μM- and 10 μM-gated currents. (**b**) *Left*, example recordings (*dashed line*, zero current baseline), and *right*, mean ± SEM (n = 3 or 5 oocytes as indicated) FMRFa concentration-dependent current amplitude ('$I/I_{max}$') with four-minute or eight-minute recovery between applications. Note how in the bottom-left recording, leak current develops after ~20 minutes due to oocyte

membrane rupture. (**c**) *Upper*, example recordings of currents gated by high FMRFa concentrations at different membrane potentials. *Lower*, mean ± SEM (n = 3 oocytes) ratio of current after five seconds ('$I_{sustained}$') to initial peak current ('$I_{transient}$'), at different potentials or with different peptide ligands. This ratio is similar at different membrane potentials and with different peptide ligands, indicating a similar level of desensitization at each potential and with each ligand. (**d**) Example recordings of currents gated by increasing concentrations of different peptide ligands.

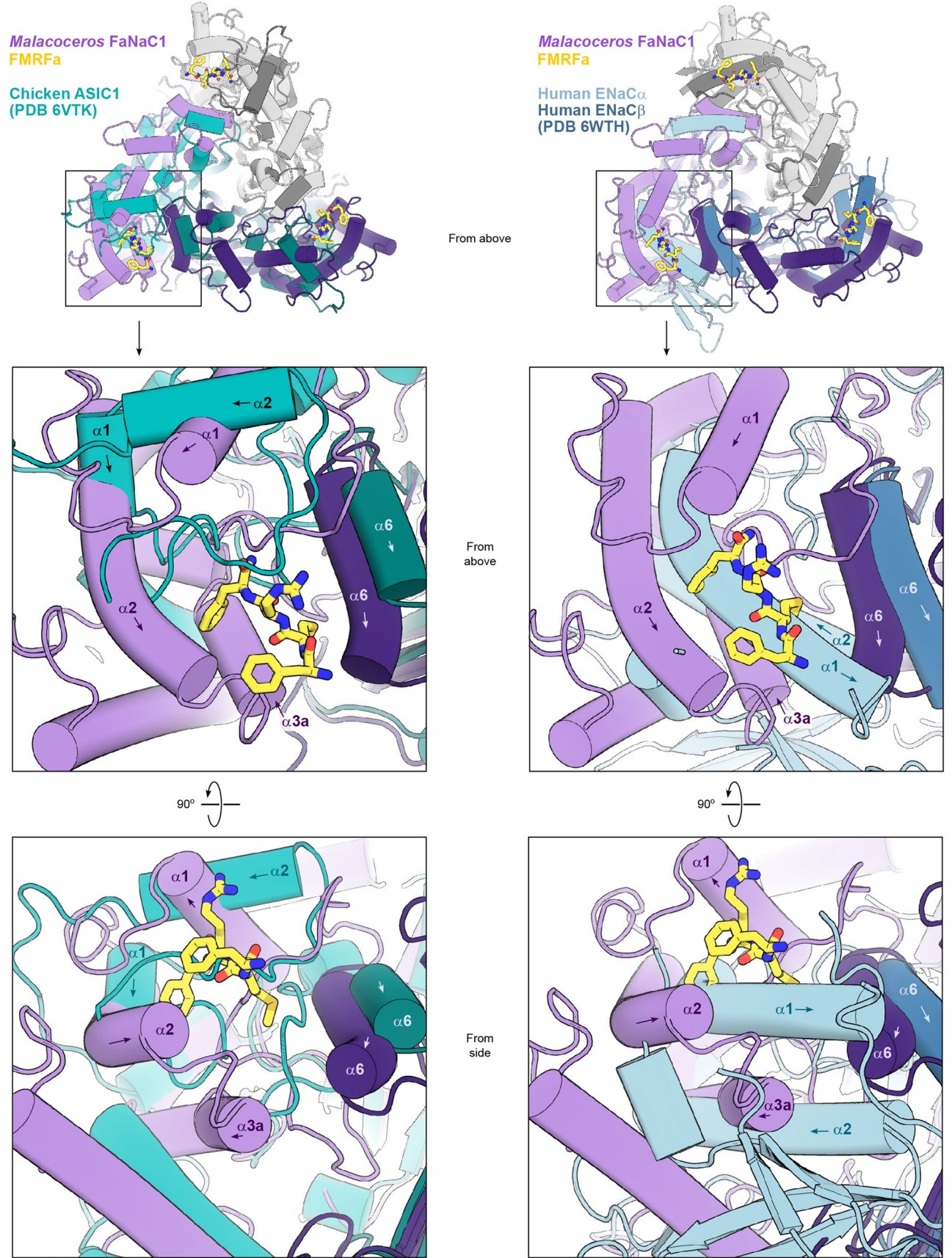

**Extended Data Fig. 6 | Comparison of FaNaC1 and other DEG/ENaC structures.** *Upper panels, Malacoceros* FaNaC1/FMRFa structure (purple and white subunits, yellow FMRFa) overlaid on two chicken ASIC1 subunits (light teal and dark teal; left) and overlaid on two human ENaC subunits (light blue and dark blue; right), viewed from the extracellular space ('from above'). *Mid panels,* magnified views of upper panels. Selected helices labelled (arrows indicate direction of peptide backbone). *Lower panels,* the same site, viewed from the side.

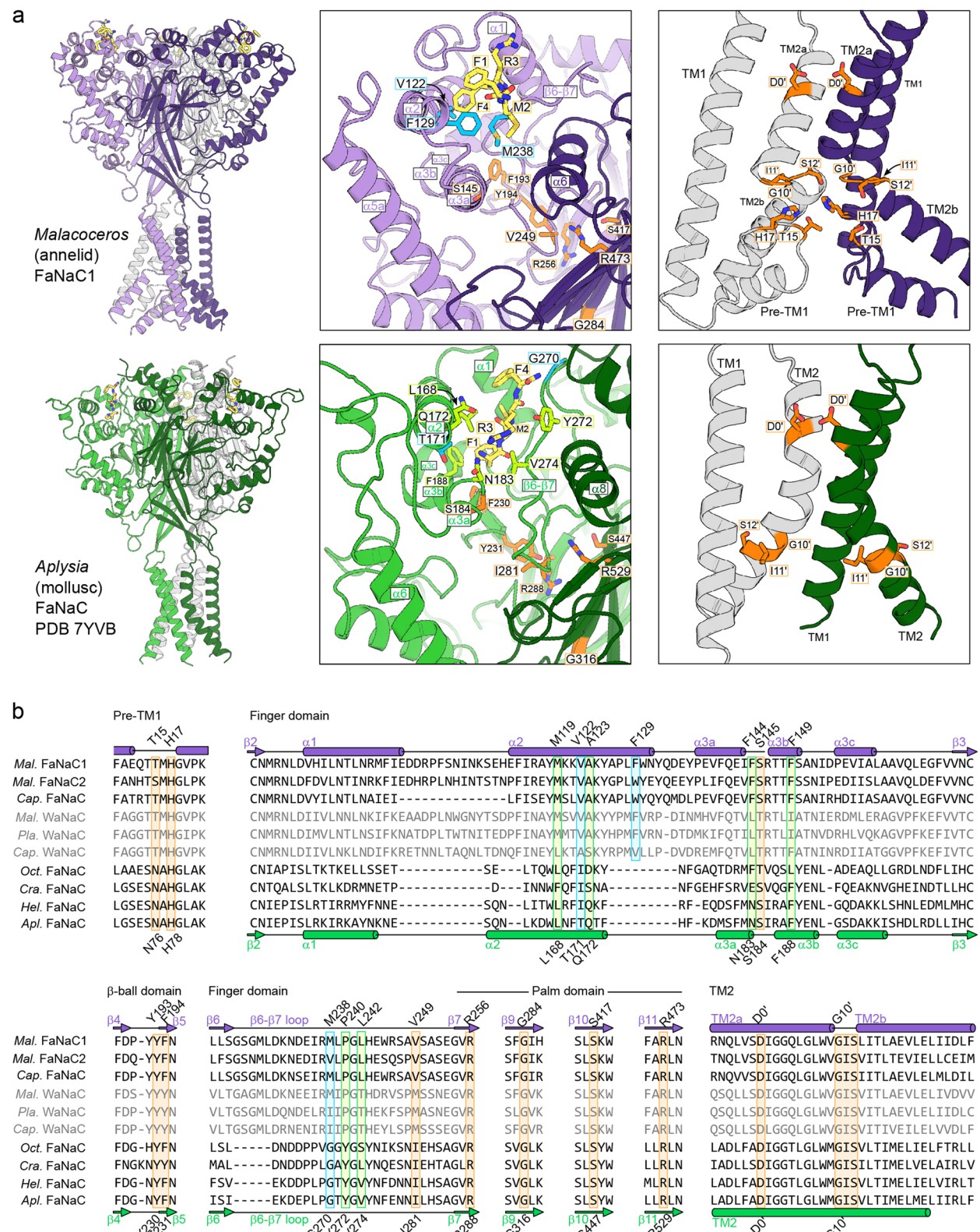

**Extended Data Fig. 7 | Comparison of *Malacoceros* FaNaC1 and *Aplysia* FaNaC.**
(**a**) Side view (left), magnified side view of FMRFa-binding site (middle), and magnified side view of channel domain with front subunit removed for clarity (right) of *Malacoceros* FaNaC1 and *Aplysia* FaNaC from elsewhere[30]. Selected *Malacoceros* FaNaC1 ligand-binding residues cyan, *Aplysia* FaNaC ligand-binding residues light green, conserved residues in interfacial site and channel domain

orange. (**b**) Amino acid sequence alignment of annelid FaNaCs (upper three; *Mal.*, *Malacoceros fuliginosus*; *Cap.*, *Capitella teleta*), annelid Wamide-gated Na⁺ channels (WaNaCs, grey; *Pla.*, *Platynereis dumerilii*), and mollusc FaNaCs (lower four; *Oct.*, *Octopus bimaculoides*; *Cra.*, *Crassostrea gigantea*; *Hel.*, *Helix aspersa*; *Apl.*, *Aplysia californica*). Secondary structure elements for *Malacoceros* and *Aplysia* FaNaCs are indicated and residues are highlighted according to (a).

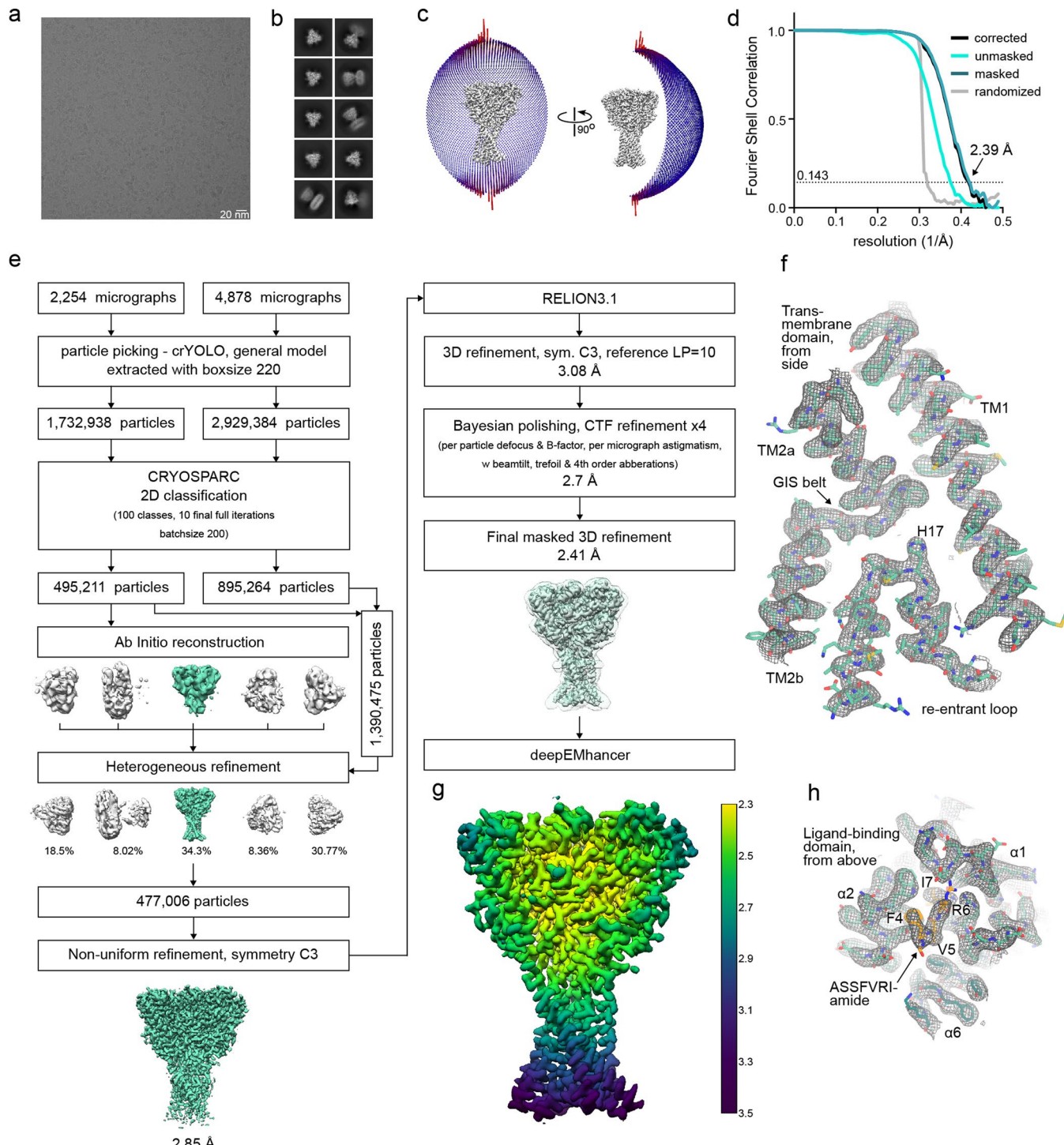

**Extended Data Fig. 8 | ASSFVRIa-bound FaNaC1 structure determination by cryo-EM.** Representative cryo-EM image of 7132 micrographs (**a**) and 2D classes (**b**) of vitrified FaNaC1 in ASSFVRIa-bound state. (**c**) Angular distribution of the particles included in the final C3-symmetrised map. The length and the colour of the sticks represent the number of particles. (**d**) FSC plot used for resolution estimation (0.143 cut-off criteria). (**e**) Schematic representation of the processing workflow, the mask used in the final refinement iteration is displayed as a transparent outline. (**f**) Density corresponding to the transmembrane domain shown as grey mesh, deepEMhancer map used for visualization is contoured at 4.5 σ. The respective fitted model is shown in teal. (**g**) Final deepEMhancer-postprocessed map coloured according to the local resolution estimation in Relion. (**h**) Close-up view of the ligand-binding site, deepEMhancer map is displayed as grey mesh contoured at 4.5 σ, fitted atomic model is displayed in teal, FMRFa molecule is shown in orange.

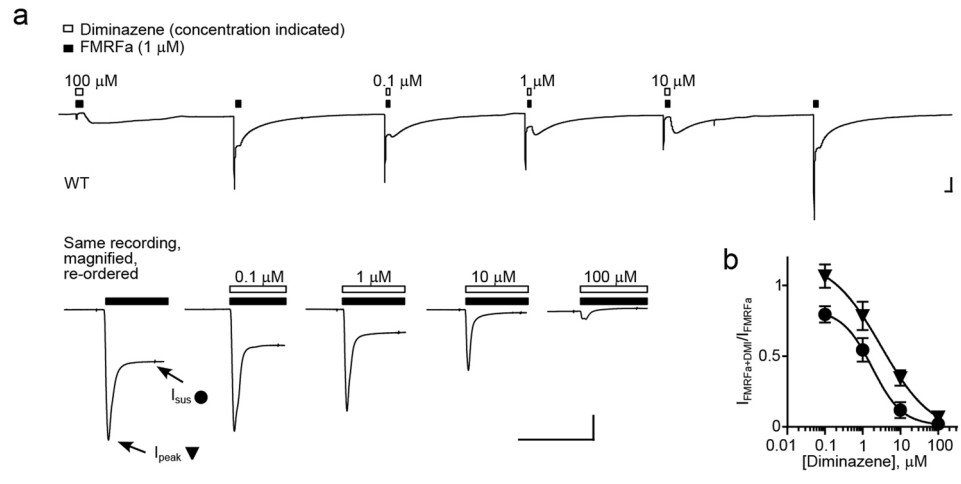

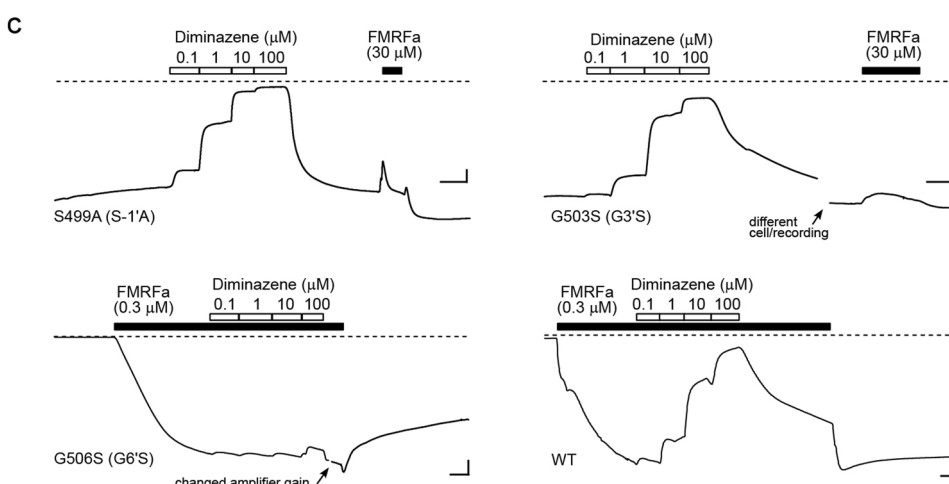

**Extended Data Fig. 9 | Effects of diminazene on mutant FaNaC1 channels. (a)** 1 μM FMRF-gated currents with and without diminazene in an oocyte expressing wildtype (WT) FaNaC1. Lower panel shows a magnified view, illustrating the inhibition of peak and sustained current by diminazene. Scale bars: x, 5 s; y, 0.2 μA. **(b)** Mean ± SEM normalized peak (n = 4 oocytes) and sustained (n = 7 oocytes) current amplitude in the presence of increasing concentrations of diminazene. **(c)** Effect of diminazene and FMRFa on constitutive currents in oocytes expressing indicated mutant FaNaC1 channels (upper panel) and effect of diminazene on sustained 0.3 μM FMRFa-gated currents in oocytes expressing mutant or WT FaNaC1 (lower panel; mean ± SEM shown in Fig. 4d). Scale bars: x, 5 s; y, 0.1 μA.

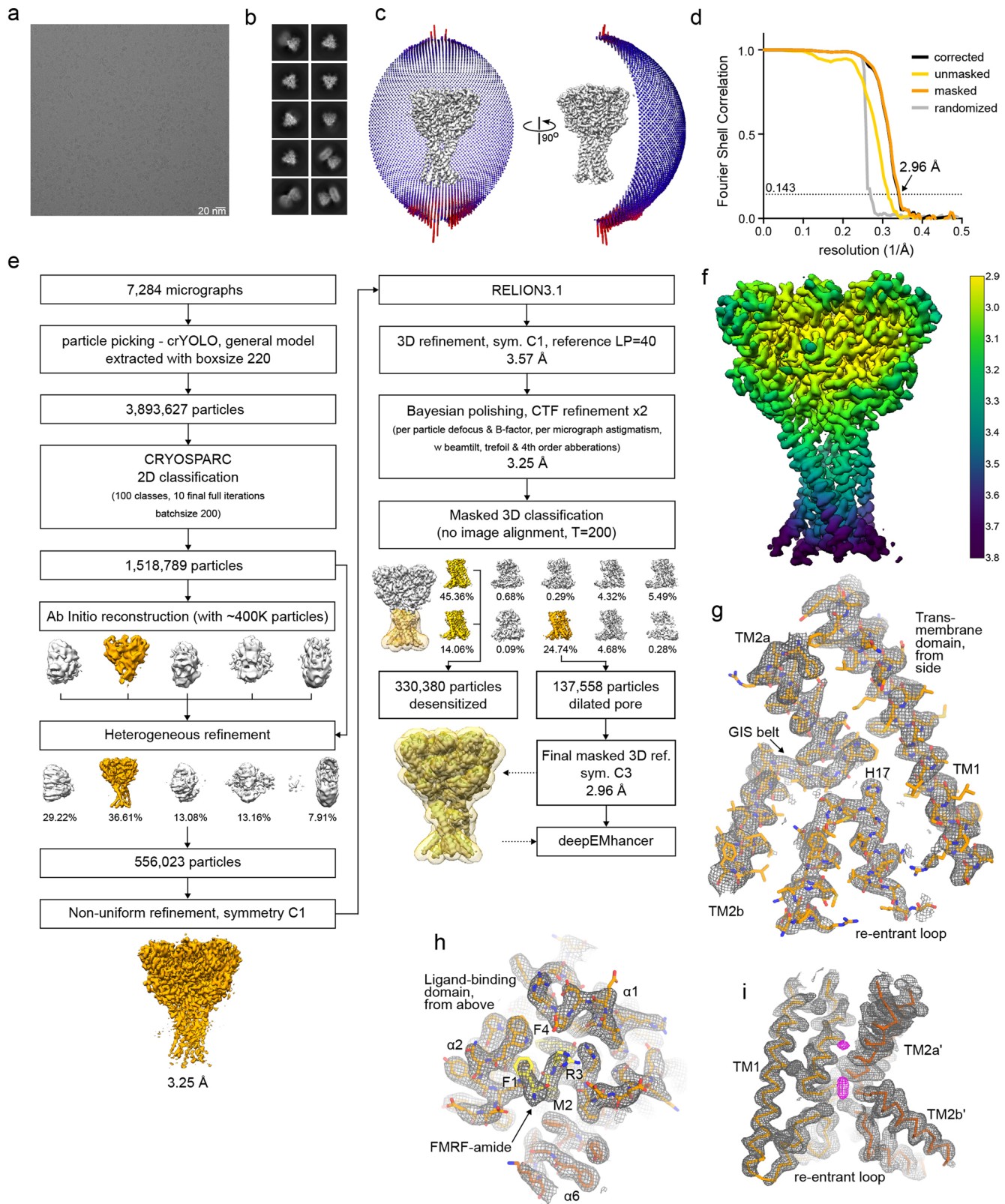

**Extended Data Fig. 10 | FMRFa-bound FaNaC1 in the presence of diminazene, structure determination by cryo-EM.** Representative cryo-EM image of 7284 micrographs (**a**) and 2D classes (**b**) of vitrified FaNaC1 in FMRFa-bound state supplemented with diminazene. (**c**) Angular distribution of the particles included in the final C3-symmetrised map. The length and the colour of the sticks represent the number of particles. (**d**) FSC plot used for resolution estimation (0.143 cut-off criteria). (**e**) Schematic representation of the processing workflow, the mask used in the final refinement iteration is displayed as a transparent outline. (**f**) Final deepEMhancer-postprocessed map coloured according to

the local resolution estimation in Relion. (**g**) Density corresponding to the transmembrane domain shown as grey mesh, deepEMhancer map used for visualization is contoured at 4 σ. The respective fitted model is shown in orange. (**h**) Close-up view of the ligand-binding site, deepEMhancer map is displayed as grey mesh contoured at 4 σ, fitted atomic model is displayed in orange, FMRFa molecule is shown in yellow. (**i**) Unassigned density in the pore region displayed in magenta, the surrounding protein density in grey, fitted atomic model is shown as Cα-trace in hues of orange. One subunit is not displayed for clarity.

# Reporting Summary

## Statistics

For all statistical analyses, confirm that the following items are present in the figure legend, table legend, main text, or Methods section.

| n/a | Confirmed | |
|---|---|---|
| ☐ | ☒ | The exact sample size (*n*) for each experimental group/condition, given as a discrete number and unit of measurement |
| ☐ | ☒ | A statement on whether measurements were taken from distinct samples or whether the same sample was measured repeatedly |
| ☒ | ☐ | The statistical test(s) used AND whether they are one- or two-sided<br>*Only common tests should be described solely by name; describe more complex techniques in the Methods section.* |
| ☒ | ☐ | A description of all covariates tested |
| ☒ | ☐ | A description of any assumptions or corrections, such as tests of normality and adjustment for multiple comparisons |
| ☐ | ☒ | A full description of the statistical parameters including central tendency (e.g. means) or other basic estimates (e.g. regression coefficient) AND variation (e.g. standard deviation) or associated estimates of uncertainty (e.g. confidence intervals) |
| ☒ | ☐ | For null hypothesis testing, the test statistic (e.g. *F*, *t*, *r*) with confidence intervals, effect sizes, degrees of freedom and *P* value noted<br>*Give P values as exact values whenever suitable.* |
| ☒ | ☐ | For Bayesian analysis, information on the choice of priors and Markov chain Monte Carlo settings |
| ☒ | ☐ | For hierarchical and complex designs, identification of the appropriate level for tests and full reporting of outcomes |
| ☒ | ☐ | Estimates of effect sizes (e.g. Cohen's *d*, Pearson's *r*), indicating how they were calculated |

*Our web collection on statistics for biologists contains articles on many of the points above.*

## Software and code

Policy information about availability of computer code

| | |
|---|---|
| Data collection | SerialEM 3.9.0 beta for cryo-EM data collection. Patchmaster 2x90.4 (HEKA) for two electrode voltage clamp. |
| Data analysis | Cryo-EM (managed through SBGrid version 2.5.6): Focus 1.1.0, crYOLO 1.8.2, MotionCor2 1.4.0, CTFfind4.1.14, cryoSPARC v3, Relion 3.1.0, deepEMhancer 220530_cu10 for cryo-EM data processing. Alphafold 2, Coot 0.9.8.1, Chimera 1.16, Phenix 1.20.1-4487, Isolde 1.6.0 for cryo-EM model building and refinement. PyMol 2.5.5, ChimeraX v 1.5 for structure visualization.<br>Clampfit v11 for two electrode voltage clamp. Prism v9 for nonlinear regression analysis. |

For manuscripts utilizing custom algorithms or software that are central to the research but not yet described in published literature, software must be made available to editors and reviewers. We strongly encourage code deposition in a community repository (e.g. GitHub). See the Nature Portfolio guidelines for submitting code & software for further information.

## Data

Policy information about availability of data

All manuscripts must include a data availability statement. This statement should provide the following information, where applicable:
- Accession codes, unique identifiers, or web links for publicly available datasets
- A description of any restrictions on data availability
- For clinical datasets or third party data, please ensure that the statement adheres to our policy

Original Malacoceros fuliginosus FaNaC1 mRNA sequence available in Genbank ON156825.1 and utilized cDNA sequence available in reference 6 (https://

## Research involving human participants, their data, or biological material

Policy information about studies with underline{human participants or human data}. See also policy information about underline{sex, gender (identity/presentation), and sexual orientation} and underline{race, ethnicity and racism}.

| | |
|---|---|
| Reporting on sex and gender | Not applicable. |
| Reporting on race, ethnicity, or other socially relevant groupings | Not applicable. |
| Population characteristics | Not applicable. |
| Recruitment | Not applicable. |
| Ethics oversight | Not applicable. |

Note that full information on the approval of the study protocol must also be provided in the manuscript.

## Field-specific reporting

Please select the one below that is the best fit for your research. If you are not sure, read the appropriate sections before making your selection.

☒ Life sciences ☐ Behavioural & social sciences ☐ Ecological, evolutionary & environmental sciences

For a reference copy of the document with all sections, see nature.com/documents/nr-reporting-summary-flat.pdf

## Life sciences study design

All studies must disclose on these points even when the disclosure is negative.

| | |
|---|---|
| Sample size | No statistical methods were used to predetermine sample size. Cryo-EM data collection was deemed sufficient if it was possible to achieve isotropic reconstructions at resolution 4Å or better for all of the datasets. For electrophysiology, experiments were generally performed on at least four different oocytes, as described in text. This is relatively standard in the field. |
| Data exclusions | Micrographs with the resolution 6Å or worse, displaying high drift, ice contamination or cracks, defocus values > 2 um and <0.3 um were excluded, according to the standard in the field. Further, particles not contributing to 2D classes with identifiable FaNaC1 features, and not contributing to the highest resolution map during 3D classification were excluded from subsequent processing. For electrophysiology, no exclusion criteria were established before experiments; obvious outliers (e.g. an oocyte showing no currents when all other oocytes in that condition showed currents) were excluded from analysis. |
| Replication | Protein was purified and reconstituted into lipid nanodiscs 3 times with similar results. Cryo-EM data collection and structure determination were performed once per dataset. The data was collected from 2 – 3 grids in all cases. For electrophysiology, all experiments were repeatable, e.g. across different oocytes, and also in different batches of oocytes. |
| Randomization | For cryo-EM, particles were randomized between even/odd groups during refinement and resolution estimation (gold-standard FSC). For electrophysiology, experiments were not performed in any particular order, so there was some "randomization", but there was no randomization in data analysis, as we were simply calculating means or pharmacological parameters. |
| Blinding | Blinding is not applicable in cryo-EM processing workflows as data are processed in an automated fashion, and for electrophysiology, we did not perform blinding, as we carefully labeled dishes of oocytes before recording from them. |

## Reporting for specific materials, systems and methods

We require information from authors about some types of materials, experimental systems and methods used in many studies. Here, indicate whether each material, system or method listed is relevant to your study. If you are not sure if a list item applies to your research, read the appropriate section before selecting a response.

## Materials & experimental systems

| n/a | Involved in the study |
|----|----|
| ☐ | ☒ Antibodies |
| ☐ | ☐ Eukaryotic cell lines |
| ☐ | ☐ Palaeontology and archaeology |
| ☐ | ☐ Animals and other organisms |
| ☐ | ☐ Clinical data |
| ☐ | ☐ Dual use research of concern |
| ☐ | ☐ Plants |

## Methods

| n/a | Involved in the study |
|----|----|
| ☐ | ☐ ChIP-seq |
| ☐ | ☐ Flow cytometry |
| ☐ | ☐ MRI-based neuroimaging |

## Antibodies

| | |
|----|----|
| Antibodies used | GFP enhancer nanobody, commonly called 3K1K |
| Validation | This is a sequenced nanobody, verified at length by Kirchhofer, A. et al. (2010) Nature Structural & Molecular Biology 17, 133-138. |

## Eukaryotic cell lines

Policy information about cell lines and Sex and Gender in Research

| | |
|----|----|
| Cell line source(s) | HEK293T (CRL-3216) and HEK293S GnTI- (CRL-3022, ATCC) from American Tissue Culture Collection. Sf9 (12659017, ThermoFisher Scientific). Xenopus laevis frog unfertilized oocytes purchased from EcoCyte Bioscience, Germany. |
| Authentication | Cell lines were not authenticated after purchase. |
| Mycoplasma contamination | All cell lines were regularly tested for Mycoplasma contamination (every 3-4 months) and were found negative. |
| Commonly misidentified lines (See ICLAC register) | No commonly identified lines (according to ICLAC register) were used in this study. |

## Palaeontology and Archaeology

| | |
|----|----|
| Specimen provenance | not applicable |
| Specimen deposition | not applicable |
| Dating methods | not applicable |

☐ Tick this box to confirm that the raw and calibrated dates are available in the paper or in Supplementary Information.

| | |
|----|----|
| Ethics oversight | not applicable |

Note that full information on the approval of the study protocol must also be provided in the manuscript.

## Animals and other research organisms

Policy information about studies involving animals; ARRIVE guidelines recommended for reporting animal research, and Sex and Gender in Research

| | |
|----|----|
| Laboratory animals | No laboratory animals used. |
| Wild animals | No wild aninamals were used. |
| Reporting on sex | Not applicable |
| Field-collected samples | Not applicable |
| Ethics oversight | Not applicable |

Note that full information on the approval of the study protocol must also be provided in the manuscript.

# Clinical data

Policy information about clinical studies
All manuscripts should comply with the ICMJE guidelines for publication of clinical research and a completed CONSORT checklist must be included with all submissions.

| | |
|---|---|
| Clinical trial registration | No clinical data used. |
| Study protocol | No clinical data used. |
| Data collection | No clinical data used. |
| Outcomes | No clinical data used. |

# Dual use research of concern

Policy information about dual use research of concern

## Hazards

Could the accidental, deliberate or reckless misuse of agents or technologies generated in the work, or the application of information presented in the manuscript, pose a threat to:

| No | Yes | |
|---|---|---|
| ☒ | ☐ | Public health |
| ☒ | ☐ | National security |
| ☒ | ☐ | Crops and/or livestock |
| ☒ | ☐ | Ecosystems |
| ☒ | ☐ | Any other significant area |

## Experiments of concern

Does the work involve any of these experiments of concern:

| No | Yes | |
|---|---|---|
| ☒ | ☐ | Demonstrate how to render a vaccine ineffective |
| ☒ | ☐ | Confer resistance to therapeutically useful antibiotics or antiviral agents |
| ☒ | ☐ | Enhance the virulence of a pathogen or render a nonpathogen virulent |
| ☒ | ☐ | Increase transmissibility of a pathogen |
| ☒ | ☐ | Alter the host range of a pathogen |
| ☒ | ☐ | Enable evasion of diagnostic/detection modalities |
| ☒ | ☐ | Enable the weaponization of a biological agent or toxin |
| ☒ | ☐ | Any other potentially harmful combination of experiments and agents |

# Plants

| | |
|---|---|
| Seed stocks | No plants used. |
| Novel plant genotypes | No plants used. |
| Authentication | No plants used. |

# ChIP-seq

## Data deposition

☐ Confirm that both raw and final processed data have been deposited in a public database such as GEO.

☐ Confirm that you have deposited or provided access to graph files (e.g. BED files) for the called peaks.

| | |
|---|---|
| Data access links<br>*May remain private before publication.* | *For "Initial submission" or "Revised version" documents, provide reviewer access links. For your "Final submission" document, provide a link to the deposited data.* |
| Files in database submission | *Provide a list of all files available in the database submission.* |
| Genome browser session<br>(e.g. UCSC) | *Provide a link to an anonymized genome browser session for "Initial submission" and "Revised version" documents only, to enable peer review. Write "no longer applicable" for "Final submission" documents.* |

## Methodology

| | |
|---|---|
| Replicates | *Describe the experimental replicates, specifying number, type and replicate agreement.* |
| Sequencing depth | *Describe the sequencing depth for each experiment, providing the total number of reads, uniquely mapped reads, length of reads and whether they were paired- or single-end.* |
| Antibodies | *Describe the antibodies used for the ChIP-seq experiments; as applicable, provide supplier name, catalog number, clone name, and lot number.* |
| Peak calling parameters | *Specify the command line program and parameters used for read mapping and peak calling, including the ChIP, control and index files used.* |
| Data quality | *Describe the methods used to ensure data quality in full detail, including how many peaks are at FDR 5% and above 5-fold enrichment.* |
| Software | *Describe the software used to collect and analyze the ChIP-seq data. For custom code that has been deposited into a community repository, provide accession details.* |

# Flow Cytometry

## Plots

Confirm that:

☐ The axis labels state the marker and fluorochrome used (e.g. CD4-FITC).

☐ The axis scales are clearly visible. Include numbers along axes only for bottom left plot of group (a 'group' is an analysis of identical markers).

☐ All plots are contour plots with outliers or pseudocolor plots.

☐ A numerical value for number of cells or percentage (with statistics) is provided.

## Methodology

| | |
|---|---|
| Sample preparation | No flow cytometry was used. |
| Instrument | No flow cytometry was used. |
| Software | No flow cytometry was used. |
| Cell population abundance | No flow cytometry was used. |
| Gating strategy | No flow cytometry was used. |

☐ Tick this box to confirm that a figure exemplifying the gating strategy is provided in the Supplementary Information.

# Magnetic resonance imaging

## Experimental design

| | |
|---|---|
| Design type | No MRI was used. |
| Design specifications | No MRI was used. |

| Behavioral performance measures | No MRI was used. |
|---|---|

## Acquisition

| Imaging type(s) | No MRI was used. |
|---|---|
| Field strength | No MRI was used. |
| Sequence & imaging parameters | No MRI was used. |
| Area of acquisition | No MRI was used. |

Diffusion MRI ☐ Used ☒ Not used

## Preprocessing

| Preprocessing software | No MRI was used. |
|---|---|
| Normalization | No MRI was used. |
| Normalization template | No MRI was used. |
| Noise and artifact removal | No MRI was used. |
| Volume censoring | No MRI was used. |

## Statistical modeling & inference

| Model type and settings | No MRI was used. |
|---|---|
| Effect(s) tested | No MRI was used. |

Specify type of analysis: ☐ Whole brain ☐ ROI-based ☐ Both

| Statistic type for inference | No MRI was used. |
|---|---|

(See Eklund et al. 2016)

| Correction | No MRI was used. |
|---|---|

## Models & analysis

| n/a | Involved in the study |
|---|---|
| ☒ | ☐ Functional and/or effective connectivity |
| ☒ | ☐ Graph analysis |
| ☒ | ☐ Multivariate modeling or predictive analysis |

