## [Peer Review File · Nature Structural & Molecular Biology]

Peer Review Information

Manuscript Title: Structural basis for excitatory neuropeptide signaling

Corresponding author name(s): Timothy Lynagh, Christina Paulino

Reviewer Comments & Decisions:

Decision Letter, initial version:

Message: 6th Jul 2023

Dear Dr. Lynagh,

Thank you again for submitting your manuscript "Structural basis for excitatory neuropeptide signaling". I apologize for the delay in responding, which resulted from the difficulty in obtaining suitable referee reports. Nevertheless, we now have comments (below) from the 3 reviewers who evaluated your paper. In light of those reports, we remain interested in your study and would like to see your response to the comments of the referees, in the form of a revised manuscript.

You will see that while all reviewers appreciate the results, they raise concerns which will need to be addressed in a revision. Specifically, we would ask that you revisit the analysis of dose response curved in line with reviewer's #1 suggestions, adding controls where necessary. As reviewer #2 notes, the binding modes of the ligands are unexpected, and would need further support, to exclude non-specific binding. We would ask you to explore further mutagenesis to validate proposed binding modes.

Please be sure to address/respond to all concerns of the referees in full in a point-by-point response and highlight all changes in the revised manuscript text file. If you have comments that are intended for editors only, please include those in a separate cover letter.

We expect to see your revised manuscript within 3 months. If you cannot send it within this time, please contact us to discuss an extension; we would still consider your revision, provided that no similar work has been accepted for publication at NSMB or published elsewhere.

Reporting Summary:

When submitting the revised version of your manuscript, please pay close attention to our [href="https://www.nature.com/nature-portfolio/editorial-policies/image-integrity">Digital Image Integrity Guidelines. and to the following points below:](https://www.nature.com/nature-portfolio/editorial-policies/image-integrity)

Please note that all key data shown in the main figures as cropped gels or blots needs to be presented in uncropped form, with molecular weight markers. These data can be aggregated into a single supplementary figure item. While these data can be displayed in a relatively informal style, they must refer back to the relevant figures.

SOURCE DATA: we request that authors provide, in tabular form, the data underlying the graphical representations used in figures. This is to further increase transparency in data reporting, as detailed in this editorial (<http://www.nature.com/nsmb/journal/v22/n10/full/nsmb.3110.html>). Spreadsheets can be submitted in excel format. Only one (1) file per figure is permitted; thus, for multi-paneled figures, the source data for each panel should be clearly labeled in the Excel file; alternately the data can be provided as multiple, clearly labeled sheets in an Excel file. When submitting files, the title field should indicate which figure the source data pertains to. We encourage our authors to provide source data at the revision stage, so that they are part of the peer-review process.

Data availability: this journal strongly supports public availability of data. All data used in

accepted papers should be available via a public data repository, or alternatively, as Supplementary Information. If data can only be shared on request, please explain why in your Data Availability Statement, and also in the correspondence with your editor. Please note that for some data types, deposition in a public repository is mandatory - more information on our data deposition policies and available repositories can be found below: <https://www.nature.com/nature-research/editorial-policies/reporting-standards#availability-of-data>

Nature Structural & Molecular Biology is committed to improving transparency in authorship. As part of our efforts in this direction, we are now requesting that all authors identified as 'corresponding author' on published papers create and link their Open Researcher and Contributor Identifier (ORCID) with their account on the Manuscript Tracking System (MTS), prior to acceptance. This applies to primary research papers only. ORCID helps the scientific community achieve unambiguous attribution of all scholarly contributions. You can create and link your ORCID from the home page of the MTS by clicking on 'Modify my Springer Nature account'. For more information please visit [visit www.springernature.com/orcid](http://www.springernature.com/orcid).

[redacted]

Sincerely,
Kat

Katarzyna Ciazynska
(she/her)
Associate Editor
Nature Structural & Molecular Biology

<https://orcid.org/0000-0002-9899-2428>

Referee expertise:

Referee #1: DEG/ENaC, physiology and cell biology

Referee #2: sensory transduction, structural biology

Referee #3: sensory transduction, cell biology

Reviewers' Comments:

Reviewer #1:

Remarks to the Author:

The authors have carried out a detailed analysis of the structure of an FMRFamide-gated DEG/ENaC family channel, FaNaC1, from the annelid *Malacoceros fuliginosus*. DEG/ENaCs are an especially interesting ion channel family, since members have a diverse range of ligands and modulators, as well as including medically important members, the Acid Sensing Ion Channels (ASICs) and Epithelial Na Channel (ENaC). The authors identify the ligand binding site and, importantly, are able to capture resting, active and desensitized states, providing a comprehensive description of conformational changes induced by ligand binding. In so doing, they provide a detailed comparison with the ASIC and ENaC channels and thus shed significant light on the commonalities and differences in their function, such as the location of ligand/tether binding sites, mechanisms of desensitization and ion permeability.

To date, our structural knowledge of the DEG/ENaC family is limited to a chicken ASIC and a human ENaC. Structural comparison with other family members, particularly with different ligands, is critical to understanding the principles of ligand recognition and gating of the DEG/ENaC channels, and the many and varied invertebrate channels represent a valuable resource with which to do this. This study is thus a timely and significant contribution to the field, providing a well-evidenced advance in our understanding of both the mechanisms and diversity of ligand binding and gating. It also provides a template for the dissection of the structural basis of channel function for other members of this diverse family. As such it is an exciting advance for DEG/ENaC researchers and of significant interest to the wider community of channel researchers, as well as being of evolutionary interest.

The manuscript does not have flaws prohibiting publication; the structures are well-supported by mutagenesis and electrophysiological data. The conclusions are original and well-supported by valid methodology and robust data, with appropriate use of statistics and treatment of uncertainties. The manuscript is well-constructed and clearly written. The abstract and introduction are appropriate and well-referenced.

Major comments

Figure 2b: At face value, several of the example traces appear not to be representative. Whereas the dose response curves in Figure 2c appear to fit a classic sigmoid curve, in b

several of the channels show a decrease in current at higher concentrations. This discrepancy requires explanation/resolution.

In several of the dose-response curves, the FMRFa control is represented as a fitted curve, without actual data. The lack of control data leads the reader to question whether the controls were carried out as part of the same experiment! Please include the actual data, so that the reader can determine the degree of variation, and/or clearly state in the legends and methods what these control curves actually represent and how variation in batches of oocytes was accounted for.

Minor comments

Line 42: "DEG/ENaCs are most highly expressed in neurons, where their gating causes depolarization due to selective cation permeability with moderate selectivity for Na⁺ over K⁺ ions". Many DEG/ENaCs are expressed in other cell types, sometimes exclusively. There is also evidence for some diversity in cation selectivity (e.g. DOI: 10.1085/jgp.202012655; DOI: 10.1113/JP283238). The references cited also do not fully support the assertions made in this sentence.

Line 88 onwards: The opening sentences seem superfluous. The authors describe their attempts to express two channels, one from Malacoceros and one from Octopus; they describe probing expression levels using a fluorescent tag (but data is "not shown") but continue only with the Malococeros channel.

Line 132: "strikingly different from the site we previously proposed" A brief discussion of this point would have been useful. It raises an important point about the caveat of mutagenesis without structural information.

Line 319: "Fig. S7" should be Fig. S4, I think?

The figure legends in the main manuscript do not always contain sufficient information to explain the figure, especially for a non-specialised audience. For example, Figure 1a needs a better description. Similarly, Figure 2a would benefit from an explanation of what views these are, i.e. that the top left is (presumably) viewed from the extracellular side; how the top right and bottom panels relate to this and the box in the top left panel. S4 would similarly benefit from a legend.

In places, the text in the figures is too small (especially when printed) or difficult to read (e.g. Figure 1d, the darker blue background is problematic).

Figure 1d "blue" and "cyan" are confusing, light/dark blue might be better. The very dark blue in c prints very poorly, it is too close to black.

There are a few small grammatical errors. Lines 81, 214, 296: "ligand-binding" should not be hyphenated; lines 450, 451: coot should have a capital C; Line 116: a/the is needed at the beginning of the line.

Reviewer #2:

Remarks to the Author:

This work by Kalienkova et al. provides novel insights into the structural basis underlying neuropeptide signaling in DEG/ENaC channels. Other members of this channel family can be activated by diverse stimuli, such as the previously characterized proton-sensing ASIC channels, suggesting that this family can transduce a variety of stimuli and raising the question of whether these diverse stimuli converge to a general model of channel of activation. To address this question, the authors identify a DEG/ENaC channel from the worm species *Malacoceros Fuliginosus* FaNaC1 that responds to neuropeptides. The authors proceed to characterize this channel structurally and functionally, in the apo state and also bound to neuropeptide ligands and an open-state blocker. Overall, the channel exhibits the same general architecture as seen in the previously described ASIC and ENaC channels. The density for the neuropeptide ligands FMRFa and ASSFVRIa is found in a site in FaNaC1 designated as the knuckle and finger domain, near the top corner of each subunit in the trimeric structure, an unexpected finding. The ligands are found to interact mainly through hydrophobic interactions with the binding site, and mutational data supports the role of a few of the proposed residues in mediating the interaction. In ligand-bound structures, the pore is found in a closed state despite the presence of activating ligands, which is likely due to the channel's characteristic desensitization behavior (it is unclear from the text if the channel desensitizes or inactivates, so I refer to the observed reduction in current as desensitization from now on just to be consistent). To circumvent the challenge of capturing the open-pore state in the context of the fast desensitization of these channels, the authors add a known open-pore blocker and proceed to structural studies. The resulting structure shows a wider pore that is somewhat consistent with Na⁺ permeability, however, the density for the blocker is tenuous. Overall, the work is rigorous, the experimental design is clear, and the structures are generally of good quality and high resolution. Below are my main concerns regarding whether the conclusions are robustly supported by the data:

Concerns:

Major:

1- The binding site contrasts sharply with previous mutational studies by the authors that predicted the ligand site to be located at the mid-extracellular interface, near the palm/wrist domains. This investigation has yet to reconcile with previous findings by the authors that both charged and polar residues are important for FMRFa potency, as opposed to their findings here which show that "the basis for ligand recognition appears to be mostly hydrophobic interactions". I would have liked to read an attempt at reconciling these findings with the previous results.

2- The ligand-bound structures are solved at relatively high concentration of ligand, and the ligand is found to interact mainly through hydrophobic interactions. Combined with the fact that the pore is not open in the presence of ligand (which is consistent with the desensitization of the channel but nevertheless obscures the results) and the fact that the proposed ligand binding site differs from the suggested site in previous studies, the overall results raise the possibility that at the high concentration of ligand used for structure determination, the binding site found is not the true ligand binding site but rather a secondary artifactual or non-specific site. The mutational data to support the finding shows that few mutations actually have an effect in the response to ligand, and the conservation of the residues around the proposed ligand binding site in other invertebrates that respond to this neuropeptide ligand is surprisingly low, which casts further doubts on the findings.

In these cases where the experiments are rigorous and the data is of good quality, but the results are still not decisive due to the challenges of the problem at hand, I would suggest to add all experimental evidence possible to support the claims. In this case, for example, I would have liked to see structure-guided amino acid substitutions on the neuropeptide FMRFa, to evaluate if the suggested binding pocket and binding mode can be directly tested using the structure as guide.

3- The authors propose that the partial agonist and full agonist operate through the same gating mechanism and differ only in affinity to the site, which they conclude from the lack of favorable pi-stacking interactions with the partial agonist and the less defined density of the partial agonist. This could be the case, however, it could be tested in single channel experiments, or direct binding assays. If untested, I would suggest removing the claim.

Minor:

1- In line 127 of the main text and accompanying figure S3, the authors state that 'the hydrophobic periphery of the channel is thinner than the membrane bilayer, such that the outer leaflet of the membrane bends to make way for hydrophilic, lateral fenestrations between adjacent subunits, creating a path for water and ions into the channel pore'. It is unclear if this is the suggested main entryway towards the channel pore, and how this observation differs (or not) from the entryway to the pore in ASIC channels. The suggestion that the membrane bends to accommodate the hydrophobicity profile of the protein should be stated carefully or with more support, as membrane bending is subject to a variety of biophysical constraints that can be modelled or quantified, but shouldn't be assumed to occur at no energetic cost.

2- The channel's electrophysiological behavior is curious, with a fast but partial reduction of current after activation and a sustained current component. The structural basis for this behavior is not addressed or commented.

Reviewer #3:

Remarks to the Author:

Kalienkova and colleagues report structures of FaNaC1 neuropeptide-gated ion channels in an unliganded, closed state, in a liganded and inactivated state, and in the presence of a pore-blocking compound that stabilizes an open state. FaNaC1 channels are remarkable examples of ionotropic neuropeptide receptors and interesting in their own right. They are also members of the Deg/ENaC family of ion channels, which play critical roles in many physiological processes and are regulated by diverse stimuli, including protons, neuropeptides and mechanostimuli. Understanding regulation of Deg/ENaC channels is, therefore, highly significant.

The study uses structural analysis and in vitro electrophysiological measurements to identify residues required for neuropeptide binding to FaNaC1 channels. The study further proposes a model for channel gating based on structures of channels stabilized in an open conformation by a pore-blocking compound. The data are clearly presented and compelling. I have only minor comments on the manuscript.

1. The authors note that the amino acid sequence of the neuropeptide-binding region of

the channel is not highly conserved between homologs. This observation warrants more discussion and perhaps some modeling to investigate mechanisms of ligand-binding by other members of the channel family.

2. Line 19 - In the abstract the authors state that the family of Deg/ENaC channels is unique with respect to diversity of gating mechanisms. This statement should be tempered - other families, e.g. K2P channels, are also regulated by diverse signals.

3. Line 61 - The authors refer to a 'more tangible transmitter' - this doesn't make sense.

4. Line 67 - The phrase 'may unfurl broad insights' is awkward,

5. Line 102 should read 'incorporated it into lipid nanodiscs'

6. Line 114 - As written it sounds as if the authors are claiming that the unstructured C-terminal tail is dispensable because it was not resolved. Please reword.

7. Line 336 - a reference has not been properly formatted.

Author Rebuttal to Initial comments

Reviewer #1:

Remarks to the Author:

The authors have carried out a detailed analysis of the structure of an FMRFamide-gated DEG/ENaC family channel, FaNaC1, from the annelid *Malacoceros fuliginosus*. DEG/ENaCs are an especially interesting ion channel family, since members have a diverse range of ligands and modulators, as well as including medically important members, the Acid Sensing Ion Channels (ASICs) and Epithelial Na Channel (ENaC). The authors identify the ligand binding site and, importantly, are able to capture resting, active and desensitized states, providing a comprehensive description of conformational changes induced by ligand binding. In so doing, they provide a detailed comparison with the ASIC and ENaC channels and thus shed significant light on the commonalities and differences in their function, such as the location of ligand/tether binding sites, mechanisms of desensitization and ion permeability.

To date, our structural knowledge of the DEG/ENaC family is limited to a chicken ASIC and a human ENaC. Structural comparison with other family members, particularly with different ligands, is critical to understanding the principles of ligand recognition and gating of the DEG/ENaC channels, and the many and varied invertebrate channels represent a valuable resource with which to do this. This study is thus a timely and significant contribution to the field, providing a well-evidenced advance in our understanding of both the mechanisms and diversity of ligand binding and gating. It also provides a template for the dissection of the structural basis of channel function for other members of this diverse family. As such it is an exciting advance for DEG/ENaC researchers and of significant interest to the wider community of channel researchers, as well as being of evolutionary interest.

The manuscript does not have flaws prohibiting publication; the structures are well-supported by mutagenesis and electrophysiological data. The conclusions are original and well-supported by valid methodology and robust data, with appropriate use of statistics and treatment of uncertainties. The

manuscript is well-constructed and clearly written. The abstract and introduction are appropriate and well-referenced.

Thank you for your careful consideration of our work and for your suggestions.

Major comments

Major point 1.) Figure 2b: At face value, several of the example traces appear not to be representative. Whereas the dose response curves in Figure 2c appear to fit a classic sigmoid curve, in b several of the channels show a decrease in current at higher concentrations. This discrepancy requires explanation/resolution.

Yes, we often noticed a decrease in current amplitude with super-saturating concentrations of ligand.

We never saw a further *increase* in current amplitude with super-saturating concentrations of ligand, and we therefore doubt there are two phases to ligand-induced *activation*. But we acknowledge that the decrease in current with high ligand concentrations is curious. We have therefore probed it further experimentally, questioning if it perhaps reflects (1) incomplete recovery from desensitization with our four-minute wait between ligand applications or (2) channel block by the cationic moiety of FMRFa, as has been suggested for mollusc FaNaCs.

The results of our new experiments, detailed in *Extended data figure 5. Measurement of FMRFa and other peptide potency*, show that (1) FaNaC1 recovery from applications of low FMRFa concentrations is complete within four minutes, whereas FaNaC1 recovery from applications of high FMRFa concentrations is incomplete even after eight minutes; (2) concentration-response experiments using a four-minute recovery time thus slightly underestimate the FMRFa EC50 (350 nM with four-minute *cf.* 760 nM with eight-minute in new experiments) but in being ~10 minutes shorter have the benefit of avoiding oocyte problems that make EC50s difficult to measure; and (3) higher concentrations induce similar desensitization at different membrane potentials and with uncharged FMRFa analogues, offering tentative evidence that FaNaC1 desensitization does not derive from channel block.

We have detailed or discussed our new experiments or explained our EC50 fits more clearly in the Figure 2 legend:

(c) Mean (\pm SEM, n = 4) normalized currents in response to increasing FMRFa concentrations at indicated mutants. Data points beyond saturating concentrations omitted for clarity. WT curve repeated from Fig. 1a. (d) Mean (\pm SEM, n = 4) normalized currents in response to increasing concentrations of different ligands at WT FaNaC1. Data points beyond saturating concentrations omitted for clarity. WT curve repeated from Fig. 1a.

in the Results:

We measured potency by establishing the half-maximal effective concentration (EC_{50}) of FMRFa during the increasing part of the concentration-response relationship, excluding a decrease in current amplitudes with higher FMRFa concentrations (Fig. 2b). Additional experiments suggested this is due to slow recovery from desensitization with high concentrations of the ligand (Extended Data Fig. 5a,b). As this decrease also occurs at positive membrane potentials and with other, uncharged peptide analogues (Extended Data Fig. 5c,d), we think it is not a sign of channel block by high concentrations FMRFa, which had been suggested for mollusc FaNaCs²⁶.

in the Methods:

In establishing EC_{50} values for activation by FMRFa, we waited four minutes between applications of increasing concentrations. As detailed in Extended Data Fig. 5a,b, this risked incomplete recovery from desensitization at high ligand concentrations, but it ensured better recordings without substantially affecting EC_{50} .

in the Discussion, also in response to **Reviewer #2 – Minor point 2**):

Despite macroscopic desensitization of FaNaC1 reflecting that of several ASICs, with fast entry to desensitization and then a noticeable sustained current³, the structural basis for desensitization may differ between FaNaCs and ASICs. The large rearrangement of the $\beta 11$ - $\beta 12$ linker during ASIC1 desensitization¹³ is not observed in FaNaC1, although we do observe a large conformational change of the FaNaC1 $\beta 5$ - $\beta 6$ loop, which is close to the $\beta 11$ - $\beta 12$ linker (Fig. 6b). We notice concentration-dependent desensitization in FaNaC1, with little and large decreases in current amplitude in low and high FMRFa concentrations, respectively, and with extremely slow washout of currents after high concentrations of FMRFa (Extended Data Figure 5). Rapid current decay and smaller current amplitudes with high ligand concentrations in FaNaC1 did not seem related to pore block by the ligand, in contrast to what had been suggested for mollusc FaNaCs^{26,30,48}.

and in a new Extended data figure 5. Measurement of FMRFa and other peptide potency.

Major point 2.) In several of the dose-response curves, the FMRFa control is represented as a fitted curve, without actual data. The lack of control data leads the reader to question whether the controls were carried out as part of the same experiment! Please include the actual data, so that the reader can determine the degree of variation, and/or clearly state in the legends and methods what these control curves actually represent and how variation in batches of oocytes was accounted for.

Indeed, we have represented the wildtype channel's FMRFa concentration-response curve as a dashed line in Figure 2c when comparing to mutant channels' FMRFa concentration-response curves; and as a dashed line in Figure 4a when comparing to other ligands' concentration-response curves. This dashed line for the wildtype is taken from the full wildtype data and SEM in Figure 1a. (More wildtype experiments are now shown in Extended Data Figure 5a,b.)

We have used this dashed line (1) for clarity, to put the focus on the other mutants' or other ligands' data and (2) so as not to repeat the same data from figure to figure. When measuring FMRFa activity at various mutants, we usually recorded one or two wildtype-expressing oocytes on the same day as the mutants to check/confirm the usual behaviour, but this was not always a complete concentration-response curve, is not shown, and we don't wish to have $n = 20$ for WT, in comparison to $n = 4$ for mutants. For Figure 4a/different ligands, we always applied 10 micromolar FMRFa to the same oocyte on which we did the other ligand concentration-response: this is already illustrated in Figure 4a, where FMRFa is clearly more potent and efficacious, and this is how the data for the other ligands were normalized in the curves in Figure 4a. But we have now tried to make this clearer for readers, and we have therefore added to the Figure 2c legend:

WT curve repeated from Fig. 1a.

to the Figure 3a legend:

(a) Left, FaNaC1 currents gated by ASSFVR1a, FVRIa, and FMRFa. Scale bars: x, 30 s; y, 250 nA. Right, average (\pm SEM, $n = 4$ or 5) ASSFVR1a- and FVRI-gated current amplitude normalized to 10 μ M FMRFa on the same cells. FMRFa curve repeated from Fig. 1A for display.

and to the Figure 6d legend:

WT curves repeated from Fig. 1a and Fig. 4a.

Minor comments

Line 42: "DEG/ENaCs are most highly expressed in neurons, where their gating causes depolarization due to selective cation permeability with moderate selectivity for Na⁺ over K⁺ ions". Many DEG/ENaCs are expressed in other cell types, sometimes exclusively. There is also evidence for some diversity in cation selectivity (e.g. DOI: 10.1085/jgp.202012655; DOI: 10.1113/JP283238). The references cited also do not fully support the assertions made in this sentence.

We've re-written this sentence to tone down the selectivity for Na⁺, broaden the expression to non-neuronal cells, and be more specific with references.

Line 88 onwards: The opening sentences seem superfluous. The authors describe their attempts to express two channels, one from Malacoceros and one from Octopus; they describe probing expression levels using a fluorescent tag (but data is "not shown") but continue only with the Malococeros channel.

We have now removed the description of the Octopus protein (from first paragraph of *Results* and from first paragraph of *Methods*) that we did not pursue.

Line 132: "strikingly different from the site we previously proposed" A brief discussion of this point would have been useful. It raises an important point about the caveat of mutagenesis without structural information.

We have now discussed this at greater length, performing new experiments that dissect the site we previously described and the new/real FMRFa binding site.

In the new experiments we generated mutant receptors with cysteine residues introduced either into our present "external/finger" site or the previously suggested "interfacial/palm" site. And we have then measured FMRFa activity at these cysteine-mutant receptors before and after labeling with a methanethiosulfonate reagent (MTSET), which should bind to the introduced cysteine. If the introduced cysteine is in the FMRFa binding site, MTSET should sterically hinder binding, offering a "real time" readout of sterics in the binding site.

We find that MTSET decreases FMRFa-gated currents via each of three introduced cysteines in the finger site and via one of two introduced cysteines in the palm site. But more importantly, we find that MTSET does this by decreasing FMRFa potency in the finger site, but without affecting FMRFa potency in the palm site.

This is shown in a new Figure 3, and described in the new Results section *Determinants of FMRFa activity in different parts of the channel*.

We had previously proposed an FMRFa-binding site ~20 Å away, at the interface of adjacent subunits' β-ball and palm domains in various FaNaCs (orange in Fig. 3a and Extended Data Fig. 7) based on amino acid sequence analysis and severe effects of mutations in this site⁶. Our high-resolution maps seem to disprove that, motivating us to experimentally compare the finger and palm/β-ball sites. To this end, we compared the effects of modification of introduced cysteine residues in both of the sites by 2-(trimethylammonium)ethylmethanethiosulfonate (MTSET), offering a readout of steric modification of the sites in "real-time". In the finger domain, MTSET modification of α1-F97C, β6/β7-M238C, and α2-F129C reduced FMRFa-gated current amplitude to approximately half in each case (Fig. 3a,b). In the palm domain, MTSET modification reduced β9-S282C currents to about half but had no effect on the β11-N475C mutant (Fig. 3a,b). This shows that both sites can be modulated by MTSET.

Differences between the sites emerged when we compared the effects of MTSET on FMRFa potency (Fig. 3c,d). MTSET modulation decreased FMRFa potency via each of the finger domain cysteine residues, shifting EC₅₀ values from 250 ± 80 nM to 4 ± 2 μM at F97C, 5 ± 1 μM to >10 μM at F129C, and 140 ± 70 nM to 580 ± 290 nM at M238C (each n = 3, Fig. 3c). In contrast, FMRFa EC₅₀ values were either unchanged or slightly decreased at palm domain S282C (360 ± 100 nM and 170 ± 40 nM) and N475C (230 ± 110 nM and 140 ± 70 nM, each n = 3, Fig. 3d). Thus, the real-time addition

of bulk to the finger domain site decreases FMRF-gated currents because of a decrease in FMRFa potency, presumably by impairing FMRFa binding. In contrast, addition of bulk to the palm/ β -ball site impairs FMRFa-gated currents without a decrease in potency, presumably by rendering many of the receptors on the oocyte surface inactive. This is consistent with the total loss of currents in annelid and mollusc FaNaCs carrying mutations at various sites in the interfacial palm/ β -ball site⁶.

Line 319: “Fig. S7” should be Fig. S4, I think?

We have changed this, also according to other revisions and figure numbers.

The figure legends in the main manuscript do not always contain sufficient information to explain the figure, especially for a non-specialised audience. For example, Figure 1a needs a better description. Similarly, Figure 2a would benefit from an explanation of what views these are, i.e. that the top left is (presumably) viewed from the extracellular side; how the top right and bottom panels relate to this and the box in the top left panel. S4 would similarly benefit from a legend.

We have elaborated in the Figure 1a legend,

Upper, example two-electrode voltage clamp current in a FaNaC1-expressing oocyte in response to FMRFa (application indicated by black bar). Mid, mean concentration-dependent FMRFa-gated current amplitudes normalized to maximum (mean \pm SEM, n = 4). Lower, FMRFa-gated (3 μ M) current amplitude at different oocyte membrane voltages in extracellular Na⁺ or K⁺-based solutions.

elaborated in the Figure 2a legend and added labels on the figure to indicate magnification and view,

(a) Ligand-binding site in FaNaC1/FMRFa structure, viewed from the extracellular space (“from above”, upper panels) and from below the FMRFa-binding site (, lower panel).

and we have added a descriptive legend to Fig. S4 (now Extended Data Figure 6):

Upper panels, Malacoceros FaNaC1/FMRFa structure (purple and white subunits, yellow FMRFa) overlaid on two chicken ASIC1 subunits (light teal and dark teal; left) and overlaid on two human ENaC subunits (light blue and dark blue; right), viewed from the extracellular space (“from above”). Mid panels, magnified views of upper panels. Selected helices labelled (arrows indicate direction of peptide backbone). Lower panels, the same site, viewed from the side.

In places, the text in the figures is too small (especially when printed) or difficult to read (e.g. Figure 1d, the darker blue background is problematic). Figure 1d “blue” and “cyan” are confusing, light/dark blue might be better. The very dark blue in c prints very poorly, it is too close to black.

We have changed to white text where there is blue background in Figure 1c.

We have changed Figure 1d legend to “ α -helices dark blue, β -strands light blue.”

We think this will print better in final resolution (initial submission was copy/pasted png image).

There are a few small grammatical errors. Lines 81, 214, 296: “ligand-binding” should not be hyphenated; lines 450, 451: coot should have a capital C; Line 116: a/the is needed at the beginning of the line.

We have made these corrections. (I.e. we have removed the hyphen in cases like “conformational changes induced by ligand binding” and kept the hyphen in cases like “ligand-binding site”.)

Reviewer #2:

Remarks to the Author:

This work by Kalienkova et al. provides novel insights into the structural basis underlying neuropeptide signaling in DEG/ENaC channels. Other members of this channel family can be activated by diverse stimuli, such as the previously characterized proton-sensing ASIC channels, suggesting that this family can transduce a variety of stimuli and raising the question of whether these diverse stimuli converge to a general model of channel of activation. To address this question, the authors identify a DEG/ENaC channel from the worm species *Malacoceros Fuliginosus* FaNaC1 that responds to neuropeptides. The authors proceed to characterize this channel structurally and functionally, in the apo state and also bound to neuropeptide ligands and an open-state blocker. Overall, the channel exhibits the same general architecture as seen in the previously described ASIC and ENaC channels. The density for the neuropeptide ligands FMRFa and ASSFVR1a is found in a site in FaNaC1 designated as the knuckle and finger domain, near the top corner of each subunit in the trimeric structure, an unexpected finding. The ligands are found to interact mainly through hydrophobic interactions with the binding site, and mutational data supports the role of a few of the proposed residues in mediating the interaction. In ligand-bound structures, the pore is found in a closed state despite the presence of activating ligands, which is likely due to the channel's characteristic desensitization behavior (it is unclear from the text if the channel desensitizes or inactivates, so I refer to the observed reduction in current as desensitization from now on just to be consistent). To circumvent the challenge of capturing the open-pore state in the context of the fast desensitization of these channels, the authors add a known open-pore blocker and proceed to structural studies. The resulting structure shows a wider pore that is somewhat consistent with Na⁺ permeability, however, the density for the blocker is tenuous. Overall, the work is rigorous, the experimental design is clear, and the structures are generally of good quality and high resolution. Below are my main concerns regarding whether the conclusions are robustly supported by the data:

Thank you for your effort in reviewing our manuscript and making these suggestions for improvement.

Concerns:

Major:

Major point 1 - The binding site contrasts sharply with previous mutational studies by the authors that predicted the ligand site to be located at the mid-extracellular interface, near the palm/wrist domains. This investigation has yet to reconcile with previous findings by the authors that both charged and polar residues are important for FMRFa potency, as opposed to their findings here which show that “the basis

for ligand recognition appears to be mostly hydrophobic interactions". I would have liked to read an attempt at reconciling these findings with the previous results.

To address this, we have generated additional mutant receptors with cysteine residues introduced either into our present "external/finger" site or the previously suggested "interfacial/palm" site. And we have then measured FMRFa activity at these cysteine-mutant receptors before and after labeling with a methanethiosulfonate reagent (MTSET), which should bind to the introduced cysteine. If the introduced cysteine is in the FMRFa binding site, MTSET should sterically hinder binding, offering a "real time" readout of sterics in the binding site.

We find that MTSET decreases FMRFa-gated currents via each of three introduced cysteines in the finger site and via one of two introduced cysteines in the palm site. But more importantly, we find that MTSET does this by decreasing FMRFa potency in the finger site, but without affecting FMRFa potency in the palm site.

This is shown in a new Figure 3, and described in the new Results section *Determinants of FMRFa activity in different parts of the channel*.

We had previously proposed an FMRFa-binding site ~20 Å away, at the interface of adjacent subunits' β-ball and palm domains in various FaNaCs (orange in Fig. 3a and Extended Data Fig. 7) based on amino acid sequence analysis and severe effects of mutations in this site⁶. Our high-resolution maps seem to disprove that, motivating us to experimentally compare the finger and palm/β-ball sites. To this end, we compared the effects of modification of introduced cysteine residues in both of the sites by 2-(trimethylammonium)ethylmethanethiosulfonate (MTSET), offering a readout of steric modification of the sites in "real-time". In the finger domain, MTSET modification of α1-F97C, β6/β7-M238C, and α2-F129C reduced FMRFa-gated current amplitude to approximately half in each case (Fig. 3a,b). In the palm domain, MTSET modification reduced β9-S282C currents to about half but had no effect on the β11-N475C mutant (Fig. 3a,b). This shows that both sites can be modulated by MTSET.

Differences between the sites emerged when we compared the effects of MTSET on FMRFa potency (Fig. 3c,d). MTSET modulation decreased FMRFa potency via each of the finger domain cysteine residues, shifting EC_{50} values from 250 ± 80 nM to 4 ± 2 μM at F97C, 5 ± 1 μM to >10 μM at F129C, and 140 ± 70 nM to 580 ± 290 nM at M238C (each $n = 3$, Fig. 3c). In contrast, FMRFa EC_{50} values were either unchanged or slightly decreased at palm domain S282C (360 ± 100 nM and 170 ± 40 nM) and N475C (230 ± 110 nM and 140 ± 70 nM, each $n = 3$, Fig. 3d). Thus, the real-time addition of bulk to the finger domain site decreases FMRFa-gated currents because of a decrease in FMRFa potency, presumably by impairing FMRFa binding. In contrast, addition of bulk to the palm/β-ball site impairs FMRFa-gated currents without a decrease in potency, presumably by rendering many of the receptors on the oocyte surface inactive. This is consistent with the total loss of currents in annelid and mollusc FaNaCs carrying mutations at various sites in the interfacial palm/β-ball site⁶.

Major point 2 - The ligand-bound structures are solved at relatively high concentration of ligand, and the ligand is found to interact mainly through hydrophobic interactions. Combined with the fact that the pore is not open in the presence of ligand (which is consistent with the desensitization of the channel but nevertheless obscures the results) and the fact that the proposed ligand binding site differs from the suggested site in previous studies, the overall results raise the possibility that at the high concentration of ligand used for structure determination, the binding site found is not the true ligand binding site but rather a secondary artifactual or non-specific site. The mutational data to support the finding shows that few mutations actually have an effect in the response to ligand, and the conservation of the residues around the proposed ligand binding site in other invertebrates that respond to this neuropeptide ligand is surprisingly low, which casts further doubts on the findings.

In these cases where the experiments are rigorous and the data is of good quality, but the results are still not decisive due to the challenges of the problem at hand, I would suggest to add all experimental evidence possible to support the claims. In this case, for example, I would have liked to see structure-guided amino acid substitutions on the neuropeptide FMRFa, to evaluate if the suggested binding pocket and binding mode can be directly tested using the structure as guide.

We think the concentration of FMRFa we have used is reasonably moderate in comparison to many other studies. We used 30 micromolar FMRFa, a concentration 36 x higher than the FaNaC1/FMRFa EC_{50} of 850 nanomolar. (In other words, two FMRFa molecules per each subunit of FaNaC1, which we prepared at ~15 micromolar based on protomer MW or ~5 micromolar based on trimer MW.) This is in comparison to e.g. recent GPCR/neuropeptide preparations with a ligand concentration 400 x higher than the EC_{50} (2 micromolar in structures *cf.* EC_{50} of 47 nM, <https://doi.org/10.1038/s41467-022-28510-6>) and nucleotide-gated channel preparations with a ligand concentration 540 x higher than the EC_{50} (2 millimolar in structures *cf.* EC_{50} of 3.7 micromolar, <https://doi.org/10.7554/eLife.39775>, <https://doi.org/10.1073/pnas.1401917111>).

Moreover, a very recent article by Liu et al (7th August 2023) reports cryoEM structures of apo and FMRFa-bound mollusc FaNaC, <https://doi.org/10.1038/s41589-023-01401-7> (our manuscript shows an annelid FaNaC). Liu et al's preparation used 1.5 millimolar FMRFa, 1800 x the EC_{50} for that FaNaC, and there they observed two binding sites: one closely overlapping with ours in the external corner of the protein; and another at the top of the channel pore, an unlikely spot for an agonist and perhaps a non-specific site due to high concentrations. Regarding the conserved and more specific external site, the major difference between Liu et al's mollusc FaNaC and our annelid FaNaC is that the four-residue FMRFa ligand is oriented with "N-terminal phenylalanine first" into the mollusc FaNaC pocket and "C-terminal phenylalanine first" into the annelid FaNaC pocket. This goes a long way to explaining the differences between mollusc and annelid FaNaCs in amino acid identity in this crucial site.

That being said, the notion of low conservation within the site and the desire to confirm our FaNaC1 FMRFa binding mode could both benefit from experimental testing. We have therefore (1) measured the activity of ligands that differ from FMRFa at the ligand's N-terminal phenylalanine, C-terminal phenylalanine, or "R3" arginine and (2) measured FMRFa activity at introduced-cysteine-mutant receptors before and after labeling our putative site with MTS reagents. These new experiments show that

- 1) The C-terminal FMRFa phenylalanine is far more important than the N-terminal phenylalanine for FMRFa potency in FaNaC1, consistent with our binding mode that places the C-terminal phenylalanine side chain deep in the ligand-binding site, interacting with the hydrophobic side chain of FaNaC F129. Secondly, FMRFa analogues FMQFa and FM(Cit)Fa show potency similar to FMRFa itself, consistent with our binding mode that has R3 pointing up and out of the site or toward hydrophilic side chains, and consistent with our mutagenesis results showing that mutation of FaNaC1 hydrophilic side chains has little effect on FMRFa potency. We have therefore added to the Results, Ligand-binding site section:

We further validated the FMRFa binding mode in our structures by testing the potency of analogous peptides differing from FMRFa only at the F1, R3, or F4 position. We replaced R3 with citrulline, isosteric but differing from arginine in a terminal amide and thus neutral moiety, or with glutamine, a substantially shorter amide side chain, yielding FMpFa and FMQFa, respectively. Compared to FMRFa, FMpFa was equally potent (EC_{50} $690 \pm$ nM, $n = 4$) and FMQFa was only 5-fold less potent (EC_{50} 4.5 ± 2 μ M, $n = 4$, Fig. 2d). This is further indicative that polar interactions between FMRFa R3 and FaNaC1 D101/E235 are unimportant and suggests that FMRFa R3 sits at the upper/outer part of the binding site and contributes little to potency. Removing the N-terminal F1 side chain also had relatively little effect, with AMRFa actually showing slightly increased potency compared to FMRFa (EC_{50} 300 ± 150 nM, $n = 3$). In contrast, removing the C-terminal F4 side chain drastically reduced potency, with FMRAa barely activating detectable currents at 100 μ M (Fig. 2d, Extended Data Fig. 5d). This supports our structural results and shows that potency derives primarily from FMRFa F4 and potentially M2 engaging FaNaC1 hydrophobic side chains.

- 2) The introduced cysteine experiments are detailed under **Reviewer #2, Major Point 1**, above.

Major point 3 - The authors propose that the partial agonist and full agonist operate through the same gating mechanism and differ only in affinity to the site, which they conclude from the lack of favorable pi-stacking interactions with the partial agonist and the less defined density of the partial agonist. This could be the case, however, it could be tested in single channel experiments, or direct binding assays. If untested, I would suggest removing the claim.

We acknowledge that the discussion of affinity and gating is speculative without further studies, and we have therefore removed it. The evidence for full and partial agonists sharing the same binding site and the

same conformational changes in the binding site is convincing, however, so we leave our conclusion here as:

Thus, the partial agonism of ASSFVR1a does not appear to derive from the induction of a different conformational change compared to FMRFa.

Minor:

1- In line 127 of the main text and accompanying figure S3, the authors state that ‘the hydrophobic periphery of the channel is thinner than the membrane bilayer, such that the outer leaflet of the membrane bends to make way for hydrophilic, lateral fenestrations between adjacent subunits, creating a path for water and ions into the channel pore’. It is unclear if this is the suggested main entryway towards the channel pore, and how this observation differs (or not) from the entryway to the pore in ASIC channels. The suggestion that the membrane bends to accommodate the hydrophobicity profile of the protein should be stated carefully or with more support, as membrane bending is subject to a variety of biophysical constraints that can be modelled or quantified, but shouldn’t be assumed to occur at no energetic cost.

We have now added “possibly” to this line in the first section of *Results* to be more moderate:

possibly creating a path for water and ions into the channel pore (Extended Data Fig. 4).

Regarding a comparison with ASIC1 or other DEG/ENaC structures, that is difficult, as most other structures were solved in detergent. The only DEG/ENaC structure in a lipid environment we know of is chicken ASIC1 in styrene maleic acid lipid nanodiscs (<https://doi.org/10.7554/eLife.56527>), however, the unmasked unsharpened maps with nanodisc density visible are not deposited, which would be required to make a direct comparison with our data.

We can only compare the hydrophobicity of FaNaC1 and ASIC1 residues, and as this is similar, we speculate one would also observe similar, slight membrane bending in ASIC1, noting the emerging picture that many membrane proteins of unrelated families deform the membrane, e.g. TMEM16s, ABC-type transporters, glutamate transporters. While membrane deformation has been strongly linked to function for some of those proteins, we are not pursuing it at length for FaNaC1 in our study. But the size of the aqueous cavity we observe in our structures derives in part from membrane bending, and we feel this deserves description, especially given the lack of available lipid-embedded DEG/ENaC structures.

2- The channel’s electrophysiological behavior is curious, with a fast but partial reduction of current after activation and a sustained current component. The structural basis for this behavior is not addressed or commented.

We have now elaborated on our discussion of this in the discussion section as follows, also referring to some new experiments shown in new Extended Data Figure 5.

Despite macroscopic desensitization of FaNaC1 reflecting that of several ASICs, with fast entry to desensitization and then a noticeable sustained current³, the structural basis for desensitization may differ between FaNaCs and ASICs. The large rearrangement of the β 11- β 12 linker during ASIC1 desensitization¹³ is not observed in FaNaC1, although we do observe a large conformational change of the FaNaC1 β 5- β 6 loop, which is close to the β 11- β 12 linker (Fig. 6b). We notice concentration-dependent desensitization in FaNaC1, with little and large decreases in current amplitude in low and high FMRFa concentrations, respectively, and with extremely slow washout of currents after high concentrations of FMRFa (Extended Data Figure 5). Rapid current decay and smaller current amplitudes with high ligand concentrations in FaNaC1 did not seem related to pore block by the ligand, in contrast to what had been suggested for mollusc FaNaCs^{26,30,48}.

Reviewer #3:

Remarks to the Author:

Kalienkova and colleagues report structures of FaNaC1 neuropeptide-gated ion channels in an unliganded, closed state, in a liganded and inactivated state, and in the presence of a pore-blocking compound that stabilizes an open state. FaNaC1 channels are remarkable examples of ionotropic neuropeptide receptors and interesting in their own right. They are also members of the Deg/ENaC family of ion channels, which play critical roles in many physiological processes and are regulated by diverse stimuli, including protons, neuropeptides and mechanostimuli. Understanding regulation of Deg/ENaC channels is, therefore, highly significant.

The study uses structural analysis and in vitro electrophysiological measurements to identify residues required for neuropeptide binding to FaNaC1 channels. The study further proposes a model for channel gating based on structures of channels stabilized in an open conformation by a pore-blocking compound. The data are clearly presented and compelling. I have only minor comments on the manuscript.

Thank you for your efforts in reviewing our manuscript and offering some suggestions for improvements.

1. The authors note that the amino acid sequence of the neuropeptide-binding region of the channel is not highly conserved between homologs. This observation warrants more discussion and perhaps some modeling to investigate mechanisms of ligand-binding by other members of the channel family.

Although we have not performed any modeling on FaNaC or other channels, in our revised manuscript we (1) note a very recent publication suggesting an overlapping binding site but different binding mode in a mollusc FaNaC (ours is an annelid FaNaC) by Liu et al <https://doi.org/10.1038/s41589-023-01401-7> and (2) performed new experiments to probe the binding mode of FMRFa in our structures.

In new experiments, we measured the activity of different peptide ligands: AMRFa, which differs from FMRFa at the ligand's N-terminal phenylalanine; FMRAa, which differs at the C-terminal phenylalanine, which we suggest buries deeply into the binding site, interacting with V122 and F129, the most important FaNaC1 residues of the binding site according to our mutagenesis; and FMpFa ("p" for citrulline, an uncharged arginine analogue) and FMQFa, which differ at the "R3" arginine, a position that our structures and mutagenesis suggest is not so important for potency.

These experiments, now in Figure 2d and Extended Data Figure 5d, show that the C-terminal FMRFa phenylalanine is far more important than the N-terminal phenylalanine for FMRFa potency in FaNaC1, consistent with our binding mode. Secondly, FMpFa and FMQFa show potency similar to FMRFa itself, again consistent with our binding mode. We have therefore added to the *Results, Ligand-binding site* section:

We further validated the FMRFa binding mode in our structures by testing the potency of analogous peptides differing from FMRFa only at the F1, R3, or F4 position. We replaced R3 with citrulline, isosteric but differing from arginine in a terminal amide and thus neutral moiety, or with glutamine, a substantially shorter amide side chain, yielding FMpFa and FMQFa, respectively. Compared to FMRFa, FMpFa was equally potent (EC_{50} $690 \pm$ nM, $n = 4$) and FMQFa was only 5-fold less potent (EC_{50} 4.5 ± 2 μ M, $n = 4$, Fig. 2d). This is further indicative that polar interactions between FMRFa R3 and FaNaC1 D101/E235 are unimportant and suggests that FMRFa R3 sits at the upper/outer part of the binding site and contributes little to potency. Removing the N-terminal F1 side chain also had relatively little effect, with AMRFa actually showing slightly increased potency compared to FMRFa (EC_{50} 300 ± 150 nM, $n = 3$). In contrast, removing the C-terminal F4 side chain drastically reduced potency, with FMRAa barely activating detectable currents at 100 μ M (Fig. 2d, Extended Data Fig. 5d). This supports our structural results and shows that potency derives primarily from FMRFa F4 and potentially M2 engaging FaNaC1 hydrophobic side chains.

Furthermore, we have now included a detailed comparison between our annelid FaNaC1 structure and the recently published mollusc FaNaC1 structure of Liu et al, illustrated in Extended Data Figure 7 and described in the section *Results - Divergence of FaNaCs from other DEG/ENaCs*, as follows:

More surprisingly, we notice substantial amino acid sequence divergence in the finger domain even within the FaNaC family. Although α 2-V122 is arguably somewhat conserved in various FaNaCs (valine in FaNaC1; valine, threonine or isoleucine in most other FaNaCs), other α 1, α 2, and α 3 residues are difficult to align, even including α 2-F129, the most influential residue for FMRFa potency in our experiments (Extended Data Fig. 7). Remarkably, despite such different sequences here, annelid and mollusc FaNaCs arrive at an architecturally similar FMRFa binding site, as revealed by the analysis of recently published structures of a FaNaC from the mollusc *Aplysia californica*³⁰. Finger domains of both *Malacoceros* FaNaC1 and *Aplysia* FaNaC comprise relatively vertical α 1 helices, horizontal α 2 and α 3a-b helices, and the β 6- β 7 loop, next to a horizontal α 6 (or equivalent α 8) helix from the adjacent subunit (Extended Data Fig. 7a). Although FMRFa binds in the same site in both channels, the peptide sits “horizontally” in *Malacoceros* FaNaC1, with F4 orienting deeply in the pocket toward α 2-V122 (Fig. 2a), but “vertically” in *Aplysia* FaNaC, now with F1 oriented most deeply into the pocket toward α 3b-F188 (Extended Data Fig. 7a)³⁰.

2. Line 19 - In the abstract the authors state that the family of Deg/ENaC channels is unique with respect to diversity of gating mechanisms. This statement should be tempered - other families, e.g. K2P channels, are also regulated by diverse signals.

We have changed “uniquely diverse” to “diverse” in the abstract as suggested, and we have also changed “especially diverse” to “rather diverse” in the introduction as follows.

The degenerin/epithelial sodium channel (DEG/ENaC) superfamily, which occurs throughout the animal kingdom, is diverse in terms of its gating stimuli...

Despite having arisen relatively recently, DEG/ENaCs are rather diverse in terms of stimuli, as the superfamily includes...

3. Line 61 - The authors refer to a 'more tangible transmitter' - this doesn't make sense.

We have changed to

larger, more canonical transmitters [than protons]

4. Line 67 - The phrase 'may unfurl broad insights' is awkward,

We have now changed "unfurl" to "offer".

5. Line 102 should read 'incorporated it into lipid nanodiscs'

We have inserted "it".

6. Line 114 - As written it sounds as if the authors are claiming that the unstructured C-terminal tail is dispensable because it was not resolved. Please reword.

We have changed to be clearer that previous experiments suggest this (and our not resolving it is simply consistent with this):

The 63-amino acid C-terminal tail was not resolved. This domain is highly variable across DEG/ENaCs, and experiments with ASICs suggest it's flexible and largely dispensable^{15,23,24}.

7. Line 336 - a reference has not been properly formatted.

We have tried to check the formatting of all references in our revised manuscript.

Additional points

We've corrected several small typos, not detailed here.

We've corrected some of the numbering in our alignment, now in Extended Data Figure 3.

We've corrected " $\alpha 3$ " to " $\alpha 2$ " in our cartoon in what is now Figure 6c.

We've changed "atop the pocket" to "near the entrance to the pocket" when describing the FMRFa F1 side chain in *Results, Ligand-binding site*.

We have updated our structural models according to RCSB/PDB requests, which has meant a very small adjustment to the orientation of the arginine ("R3") side chain of FMRFa and no changes to the other side chains of FMRFa. This does not change any of our interpretations, and we have updated all of the models in our figures.

New model calculations mean we no longer needed the previous ligand modeling and we have therefore deleted the following from the section *Methods, Model building and refinement and pore analysis*

The ligand coordinates were generated in Chem3D 18 (PerkinElmer), and the restraints for the ligands were generated in PRODRG⁷¹.

To explain our methods more clearly, we added the following to the section *Methods, Image processing*

Resolution was estimated in Relion-3.1 postprocessing, using a mask excluding nanodisc density according to the standard Fourier shell correlation cutoff of 0.143 (Extended Data Figures 1,2,8,10).

We've changed our previous Supplementary Information to be 10 Extended Data Figures and one Supplementary Information – Table S1.

Decision Letter, first revision:

Message: Our ref: NSMB-A47642A

25th Sep 2023

Dear Dr. Lynagh,

Thank you for submitting your revised manuscript "Structural basis for excitatory neuropeptide signaling" (NSMB-A47642A). It has now been seen by the original referees and their comments are below. The reviewers find that the paper has improved in revision, and therefore we'll be happy in principle to publish it in Nature Structural & Molecular Biology, pending minor revisions to satisfy the referees' final requests and to comply with our editorial and formatting guidelines.

Sincerely,

Katarzyna Ciazynska
(she/her)
Associate Editor
Nature Structural & Molecular Biology
<https://orcid.org/0000-0002-9899-2428>

Reviewer #1 (Remarks to the Author):

I thank the authors for their revisions, which have adequately addressed all of my comments. I have no further comments.

Reviewer #2 (Remarks to the Author):

I thank the authors for the careful revision of their manuscript, which I believe is ready for publication as is.

Reviewer #3 (Remarks to the Author):

The revised manuscript fully addresses my comments and incorporates new data that strengthen what was already a clear and compelling study.

Author Rebuttal, first revision:

Thank you for taking the time to re-review our manuscript.

Below, in blue text, we respond to the reviewers' latest comments.

Timothy Lynagh 2nd November 2023.

Reviewer #1:

Remarks to the Author:

I thank the authors for their revisions, which have adequately addressed all of my comments. I have no further comments.

Great.

Reviewer #2:

Remarks to the Author:

I thank the authors for the careful revision of their manuscript, which I believe is ready for publication as is.

Great.

Reviewer #3:

Remarks to the Author:

The revised manuscript fully addresses my comments and incorporates new data that strengthen what was already a clear and compelling study.

Great.

Final Decision Letter:

Message 5th Dec 2023

:
Dear Dr. Lynagh,

We are now happy to accept your revised paper "Structural basis for excitatory neuropeptide signaling" for publication as an Article in Nature Structural & Molecular Biology.

Your paper will be published online soon after we receive proof corrections and will appear in print in the next available issue. You can find out your date of online publication by contacting the production team shortly after sending your proof corrections. Content is published online weekly on Mondays and Thursdays, and the embargo is set at 16:00 London time (GMT)/11:00 am US Eastern time (EST) on the day of publication. Now is the time to inform your Public Relations or Press Office about your paper, as they might be interested in promoting its publication. This will allow them time to prepare an accurate and satisfactory press release. Include your manuscript tracking number (NSMB-A47642B) and our journal name, which they will need when they contact our press office.

About one week before your paper is published online, we shall be distributing a press release to news organizations worldwide, which may very well include details of your work. We are happy for your institution or funding agency to prepare its own press release, but it must mention the embargo date and Nature Structural & Molecular Biology. If you or your

Press Office have any enquiries in the meantime, please contact press@nature.com.

Please note that *Nature Structural & Molecular Biology* is a Transformative Journal (TJ). Authors may publish their research with us through the traditional subscription access route or make their paper immediately open access through payment of an article-processing charge (APC). Authors will not be required to make a final decision about access to their article until it has been accepted. [Find out more about Transformative Journals](https://www.springernature.com/gp/open-research/transformative-journals)

Sincerely,

Katarzyna Ciazynska, PhD
(she/her)
Associate Editor
Nature Structural & Molecular Biology
<https://orcid.org/0000-0002-9899-2428>
